Manuscript prepared for Atmos. Chem. Phys.
with version 2014/09/16 7.15 Copernicus papers of the LaTeX class copernicus.cls.
Date: 8 May 2017

# Disk and circumsolar radiances in the presence of ice clouds

Päivi Haapanala[1], Petri Räisänen[2], Greg M. McFarquhar[3], Jussi Tiira[1],
Andreas Macke[4], Michael Kahnert[5,6], John DeVore[7], and Timo Nousiainen[2]

[1]Department of Physics, University of Helsinki, P.O.Box 48, FI-00014 University of Helsinki, Finland
[2]Finnish Meteorological Institute, P.O.Box 503, FI-00101 Helsinki, Finland
[3]Department of Atmospheric Science, University of Illinois at Urbana-Champaign, Urbana, 105 S. Gregory ST., IL 61801-3070, USA
[4]Leibniz Institute for Tropospheric Research, Permoserstraße 15, 04318 Leipzig, Germany
[5]Research Department, Swedish Meteorological and Hydrological Institute, Folkborgsvägen 17, 601 76 Norrköping, Sweden
[6]Department of Earth and Space Science, Chalmers University of Technology, 412 96 Gothenburg, Sweden
[7]Visidyne, Inc., 429 Stanley Drive, Santa Barbara, CA 93105, USA

*Correspondence to:* Päivi Haapanala (paivi.haapanala@helsinki.fi)

**Abstract.** The impact of ice clouds on solar-disk and circumsolar radiances is investigated using a Monte Carlo radiative transfer model. The monochromatic direct and diffuse radiances are simulated at angles of 0 to 8° from the center of the sun. Input data for the model are derived from measurements conducted during the 2010 Small Particles in Cirrus (SPARTICUS) campaign together with state-of-the-art databases of optical properties of ice crystals and aerosols. For selected cases, the simulated radiances are compared with ground-based radiance measurements obtained by the Sun and Aureole Measurement (SAM) instrument.

First, the sensitivity of the radiances to the ice cloud properties and aerosol optical thickness is addressed. The angular dependence of the disk and circumsolar radiances is found to be most sensitive to assumptions about ice crystal roughness (or, more generally, non-ideal features of ice crystals) and size distribution, with ice crystal habit playing a somewhat smaller role. Second, in comparisons with SAM data, the ice-cloud optical thickness is adjusted for each case so that the simulated radiances agree closely (i.e., within 3 %) with the measured disk radiances. Circumsolar radiances at angles larger than $\approx 3°$ are systematically underestimated when assuming smooth ice crystals, whereas the agreement with the measurements is better when rough ice crystals are assumed. Our results suggest that it may well be possible to infer the particle roughness directly from ground-based SAM measurements. In addition, the results show the necessity of correcting the ground-based measurements of direct radiation for the presence of diffuse radiation in the instrument's field of view, in particular in the presence of ice clouds.

# 1 Introduction

The portion of solar radiation that appears to originate from a small disk around the sun is called circumsolar radiation or solar aureole. This radiation arises from near-forward scattering of direct solar radiation by atmospheric particles with sizes comparable to or larger than the wavelength (i.e., larger than 1 μm); the larger the particle compared to the wavelength of radiation is, the more peaked the scattering phase function $P_{11}$ is and the more scattering is concentrated at near-forward angles. Consequently, the amount of circumsolar radiation varies widely depending on the geographical, seasonal and diurnal variation of airborne particles (Norrig et al., 1991; Neuman et al., 2002). As ice crystals are typically much larger than aerosol particles or gas molecules, a considerably larger part of the direct solar radiation is scattered into the circumsolar region in the presence of ice clouds. In addition to the phase function, the amount of circumsolar radiation depends on the single-scattering albedos and extinction coefficients of atmospheric gases and particles. All these optical properties depend on wavelength. Furthermore, the ensemble/volume-averaged optical properties depend on the concentration, composition and size-shape distribution of the particles. Although the impact of ice crystal sizes and shapes on their optical properties has been studied in much detail (Macke et al., 1998; Zhang et al., 1999; McFarquhar et al., 2002; Schlimme et al., 2005; Um and McFarquhar, 2007, 2009, 2011, 2013), there is no detailed information on how ice crystals affect the angular dependence of circumsolar radiances. However, the studies of Segal-Rosenheimer et al. (2013) and Reinhardt et al. (2014) have revealed that differences in the modeled forward scattering of smooth and roughened ice crystals as well as different shape distributions of ice crystals lead to differences in the circumsolar radiation. DeVore et al. (2012) also noted the impact of ice crystals properties (roughness and effective radius) on calculated circumsolar radiances.

Circumsolar radiation is widely detected by instruments measuring the direct radiation (i.e. pyrheliometers) and therefore counted as direct radiation. Such instruments often have a half-opening angle of 2–3°, whereas the half-width of the solar disk is only about 0.27° when observed from the Earth. Depending on the ambient atmospheric conditions, the near-forward scattered radiation can be a large portion of the total radiation measured by these instruments, leading to overestimation of the amount of direct solar radiation. Therefore, retrievals of ice cloud optical thickness and other properties from the direct radiation measurements can be biased. There have been some efforts to quantify the amount of circumsolar radiation in the measured direct radiation and to account for its impact on the underestimation of cloud optical thickness (Shiobara and Asano, 1994; Kinne et al., 1997; Segal-Rosenheimer et al., 2013). For example Segal-Rosenheimer et al. (2013) proposed a new approach to derive ice cloud optical thickness and effective diameter from sun photometry measurements by using ice-cloud optical property models.

Since the circumsolar radiance distribution is usually nearly radially symmetric around the sun, it is reasonable to describe it as a function of the angular position relative to the centre of the sun (Blanc et al., 2013). This solar radiance profile is also called the sunshape. Wilbert et al. (2012) pre-

sented a method for determining the sunshape using a pair of pyrheliometers with different opening angles. The amount of circumsolar radiance and the radial profile of sunshape can also be measured using a Sun and Aureole Measurement (SAM) instrument. It consists of two solar tracking cameras: one observing the Sun disk and another the aureole. The cameras are filtered into the $670 \pm 5$ nm wavelength band. SAM measures the disk and circumsolar radiances with a very high dynamic range and produces the disk and aureole radiances as a function of angle from the center of the Sun out to $8°$ with an angular resolution of $0.0148°$. DeVore et al. (2009) demonstrated the ability of SAM measurements to derive the effective radius and optical thickness of ice clouds and DeVore et al. (2012) used MODIS retrievals of thin cirrus to calculate solar disk and aureole measurements that were compared with SAM measurements. Reinhardt et al. (2014) have also developed a method to determine circumsolar radiation from satellite observations. They noted that the uncertainties in their retrieval due to assumptions on the ice particle shape can sum up to 50 %, and even larger errors are expected if instantaneous values are compared against SAM measurements. DeVore et al. (2012) suggested that a collection of SAM measurements might provide a useful template for helping to derive phase functions of ice crystals.

There have been some efforts to account for the impact of circumsolar radiation and sunshape on concentrating solar energy applications (Bui and Monger, 2004; Reinhardt, 2013; Reinhardt et al., 2014; Wilbert et al., 2012, 2013). These applications use concentrating solar collectors whose half opening angles are typically less than $1°$. Due to the $1–2°$ smaller acceptance angle than that of a pyrheliometer, these collectors are able to use only a fraction of the circumsolar radiation measured with a pyrheliometer. Consequently, if the performance of the solar concentrating system is predicted based on measurements of direct radiation (including circumsolar radiation), the energy contained in the circumsolar region at angles from 1 to $3°$ can lead to overestimation of the performance. To better estimate and optimize the amount of received energy of the concentrating solar energy systems, the detailed angular distribution of the circumsolar radiation and how it varies in time and location should be known.

The overarching goal of this research is to understand how ice clouds influence the downwelling solar radiances within a few degrees from the direction of the sun. This knowledge could be exploited, in future work, for developing schemes to correct measurements of direct solar radiation for the diffuse radiation that is present at the angular range of instruments such as pyrheliometers. Furthermore, it is crucial for understanding the information content in measurements with the relatively new SAM instrument, and for the future development of retrieval algorithms based on SAM data. This study is largely divided into two components, both of which contribute to the overarching goal. First the parameters that the circumsolar radiance is sensitive to are identified. In particular, the impacts on circumsolar radiances due to ice crystal size-shape distribution and roughness, ice cloud optical thickness and aerosol optical thickness are simulated. For this purpose, a forward Monte Carlo radiative transfer model is used. Monochromatic downwelling radiances for various ice-cloud

scenarios are simulated at a wavelength of 0.670 μm. These scenarios are based on in-situ-measured size distributions of mid-latitude ice clouds together with either measurement-based shape distributions or idealized single-habit distributions. These size-shape distributions of ice crystals are combined with a database of single-scattering properties of ice crystals (Yang et al., 2013) to produce size-shape-integrated bulk optical properties of the ice clouds as needed for input to the radiative transfer model. The in-situ-based distributions of ice crystals were obtained from aircraft measurements made over the Atmospheric Radiation Measurement (ARM) program's Southern Great Plains (SGP) site (36.606° N, 97.485° W) during the Small Particles in Cirrus (SPARTICUS) campaign conducted in 2010. In the second part of this work, case studies are used to determine the degree of agreement between selected ground-based solar-disk and circumsolar radiance measurements by the SAM instrument at the SGP site and simulated radiances. It should be noted that the first component (sensitivity studies) provides important information for designing and interpreting the comparison with modeled radiances in addition to providing a fundamental understanding on how ice crystal properties affect circumsolar radiance.

## 2 Radiative transfer model

In this study, the angular dependence of solar disk and circumsolar radiances are simulated with a modified version of the Monte Carlo Model of the University of Kiel (MC-UniK) developed by Macke et al. (1999). Even though a plane-parallel horizontally homogeneous atmosphere is assumed in the radiation calculations (see below), the Monte Carlo technique is applied because of its flexibility. Specifically, it allows a consideration of the finite width of the sun and the computation of radiances at an arbitrarily high angular resolution in the vicinity of the direction of the Sun, without incurring extreme computational costs. In fact, we are not aware of any deterministic radiative transfer models that would satisfy these criteria.

### 2.1 Technical details

The MC-UniK is a forward Monte Carlo model for efficient calculations of radiances at discrete directions. It employs the Local Estimate Method (e.g., Marshak and Davis, 2005) and has been validated within the Intercomparison of 3-D-Radiation Codes project (Cahalan et al., 2005). The model simulates the scattering events of photons within the ice cloud/atmosphere using a non-truncated treatment for the phase functions. The free path length is based on Beer's law and gives the distance between two successive scattering processes. The scattering direction is derived using a random number generator so that the scattering angle $s$ corresponding to a given random number [0,1] equals the cumulative phase function from 0 to $s$, and the azimuth angle is sampled uniformly in the range $[0, 2\pi]$. Absorption is taken into account by multiplying the photon weight by the local single scattering albedo. For reasons of variance reduction and computing time, techniques as proposed by

Barker et al. (2003) have been implemented. For calculating the radiance field, the Local Estimate Method is more efficient than the common Monte Carlo photon counting method because no photons get lost. Thus, in effect, MC-UniK assumes that a fraction of the photon is scattered directly into each detector. These photons are attenuated along the optical path between the scattering location and the detector.

## 2.2 Modifications

We have modified the original MC-UniK to account for the finite width of the solar disk, that is an opening angle of $0.534°$. In addition, a phenomenon called limb darkening is accounted for. The solar radiation that reaches the observer originates in the photosphere of the Sun peaking at an optical depth of roughly unity along the line of sight. On average, this corresponds to a temperature of about 5778 K. However, along a slant line of sight toward the Sun's limb, an optical depth of one is reached at a higher altitude with a lower temperature. Hence the intensity reaching the observer from the limb of the Sun is lower than that from the center (Green and Jones, 2015). In our version of MC-UniK the limb darkening is taken into account by using the formula

$$I(\beta) = I(0,0)[a + b\cos(\beta) + c\cos^2(\beta)] \tag{1}$$

given in Böhn-Vitense (1989), where $\beta$ is the angular distance from the center of the Sun to the limb $(0°–90°)$. At $\lambda=0.69\,\mu m$ (the closest available wavelength to $0.67\,\mu m$), the coefficients have values of $a = 0.4128$, $b = 0.7525$, and $c = -0.1761$.

The model output is modified to include the direct and diffuse radiances at the surface (in units of $Wcm^{-2}\mu m^{-1}sr^{-1}$) for specified detector positions. For the mean solar constant at $\lambda=0.670\,\mu m$, values of $0.1509\,Wcm^{-2}\mu m^{-1}$ (Gueymard, 2004) and $2206\,Wcm^{-2}\mu m^{-1}sr^{-1}$ are used in the calculation of diffuse and direct radiances, respectively. The latter value is obtained by dividing the former by the solid angle of the sun.

## 2.3 Input

The model domain is separated into grid boxes which are characterized by their bulk optical properties: the volume extinction coefficient $K_{ext}$, the single-scattering albedo $\omega$, and the scattering phase function $P_{11}(\gamma)$, where $\gamma$ is the scattering angle. Here the model domain of MC-UniK is divided into 15 non-uniform vertical layers extending from the ground up to 50 km. Gas absorption and Rayleigh scattering occur in all layers, while aerosols are assumed to be confined to the lowest layer below 2 km. The ice cloud resides between 8.0 and 11.5 km (model layers $5-11$) depending on the case (see Sect. 3.2). A plane parallel cloud is assumed due to insufficient information on the cloud horizontal structure. Thus, while the Monte Carlo model can account for 3D effects, the effects related to cloud horizontal inhomogeneity are not accounted for.

Furthermore, the solar zenith angle ($\theta$), detector positions, and surface albedo data are required. A total of 418 detectors pointing to the Sun and its surrounding areas inside the opening angle of $16°$ are positioned so that they cover both the horizontal and vertical cross sections of the area as illustrated in Fig. 1. For surface albedo, a fixed value of 0.2 is used. To achieve sufficient accuracy for the calculations, 8 million photons are used. At the angles considered here (0–8° from the center of Sun), the resulting random errors are mostly below 3 % (6 %) for rough (smooth) crystals, with smaller errors at the smaller angles.

## 3   Optical properties

The optical properties of ice clouds (and atmospheric gases and aerosols) needed as input to the MC-UniK are based on data collected during the Atmospheric Radiation Measurement program's 2010 Small Particles in Cirrus (SPARTICUS) field campaign (Mishra et al., 2014; Muhlbauer et al., 2014; Jackson et al., 2015). The aircraft measurements were collected in the vicinity of the ground-based measurements made at the SGP site. Out of the numerous case days of SPARTICUS, only two were deemed suitable for the present investigation: 23 March (hereafter flight A) and 24 June (hereafter flight B). During these flights, there was a visually observable cirrus cloud without lower cloud layers and all the needed in situ and ground-based measurement data had good quality.

### 3.1   Optical properties of atmospheric gases and aerosols

To account for Rayleigh scattering and gas absorption, the optical properties ($\omega$ and $K_{ext}$) of the atmosphere without cloud and aerosols are calculated using the scheme of Freidenreich and Ramaswamy (1999). The spectral band of 0.599–0.685 μm is used for gas absorption, with Rayleigh scattering optical depth scaled to 0.67 μm. The vertical profiles of temperature and water vapour are based on radiosondes launched at the SGP site during the case days, complemented by ERA-Interim reanalysis data (Dee et al., 2011) in the middle and upper stratosphere. Ozone profiles are taken from the ERA-Interim data. The phase function for Rayleigh scattering is $P_{11}(\gamma) = (3/4)(1 + \cos^2 \gamma)$.

The ensemble-averaged aerosol $\omega$ and $P_{11}(\gamma)$ are taken from the OPAC (Optical Properties of Aerosols and Clouds) database (Hess et al., 1998), assuming values for continental average aerosols at $\lambda = 0.650$ μm computed at a relative humidity of either 70 % (for comparison with SAM measurements during flight B) or 50 % (for all other calculations).The aerosol optical thickness $\tau_a$ is estimated from the AERONET level 1.5 $\tau_a$ retrieval (at $\lambda = 0.675$ μm) and from the visible Multifilter Rotating Shadowband Radiometer (MFRSR) measurements (at $\lambda = 0.673$ μm) conducted at the SGP site, which yields $\tau_a = 0.09$ during flight A and $\tau_a = 0.166$ during flight B. The aerosol $K_{ext}$ is derived from $\tau_a$ assuming that the aerosols are confined to the lowest 2 km.

### 3.2 Ice crystal size-shape distributions

During SPARTICUS in situ probes were installed on the Stratton Park Engineering Company (SPEC) Inc. Learjet 25 aircraft. The Learjet conducted 101 missions sampling several cirrus clouds in the mid-latitudes of the United States at temperatures between $-70$ and $-20^\circ$ C. Jackson et al. (2015) examined all the size and shape distributions sampled by the SPEC Learjet during SPARTICUS, establishing the meteorological context of each cirrus sampled. The two flights analyzed here are unique in that the cirrus sampled had no underlying cloud layers below. The probes on the Learjet that are used in this study include the Cloud Particle Imager (CPI) acquiring high 2.3 μm resolution images of particles, the Fast Forward Scattering Spectrometer Probe (FFSSP) measuring particles with maximum diameter ($D_{max}$) smaller than 50 μm from the forward scattering of light, the Two-Dimensional Stereo (2DS) probe nominally measuring particles with $10 < D_{max} < 1280$ μm, and a 2-D Precipitation Probe (2DP) measuring particles with $200 < D_{max} < 6400$ μm for flight A and a High Volume Precipitation Sampler (HVPS-3) measuring particles with $150 < D_{max} < 19200$ μm for flight B.

The composite size distributions required to calculate the microphysical and optical properties were determined using the FFSSP to characterize particles with $D_{max} < 50$ μm, the 2DS for 50 μm $< D_{max} < 1200$ μm, and the 2DP or HVPS-3 for larger particles. Concentrations of small ice crystals (defined as those with $D_{max} < 100$ μm) are, however, highly uncertain due to a small and poorly defined sample volume (Baumgardner and Korolev, 1997; McFarquhar et al., 2016) and potential contributions from shattered artifacts (e.g. Gardiner and Hallett, 1985; McFarquhar et al., 2007; Korolev et al., 2011, 2013) in both the 2DS and FFSSP. Therefore, four alternative representations of the concentration of small ice crystals are used to test the sensitivity of the results to these concentrations. In $small_{100\%}$, the concentration of crystals with $D_{max} < 100$ μm is taken directly from the FFSSP and 2DS measurements. In $small_{0\%}$, $small_{50\%}$ and $small_{200\%}$ the measured concentration is multiplied by 0 (i.e., no small ice crystals), 0.5 and 2, respectively.

For large ice crystals ($D_{max} > 100$ μm), the size-dependent shape distributions are based on the CPI images measured in situ. Um and McFarquhar (2011) and Ulanowski et al. (2004) show that the detailed shapes of small ice crystals cannot be identified using the CPI due to its limited image resolution and blurring of images due to diffraction that renders the shape classification of small ice crystals unreliable. Due to the lack of reliable in situ measurements of the shapes of crystals with $D_{max} < 100$ μm, they are assumed to be hollow columns. For large crystals, an automatic ice-cloud particle habit classifier IC-PCA (Lindqvist et al., 2012) is used to determine the fraction of different habits as a function of particle size from the CPI images. The IC-PCA automatically sorts the crystals into 8 classes: bullet, column, column aggregate, bullet rosette, bullet rosette aggregate, plate, plate aggregate, and irregular. In this study bullets are classified as columns and bullet rosette aggregates as column aggregates due to the lack of information about their single-scattering properties. The final six habit classes listed in Table 1 are named as column, column agg, bullet rosette, plate, plate

agg, and irregular. The size-resolved shape distributions are created by combining the size distributions (measured by 2DS and 2DP or HVPS-3) and the relative portions of the size-resolved shape distributions from CPI/IC-PCA at each layer.

Based on the stepwise flight path of the aircraft, the measurements of ice crystals are sorted into 0.5 km vertical layers. In each layer, the particle concentrations and size distributions are averaged
over the time the Learjet was in the appropriate layer. During flight A the cloud was present in four of the layers (from 9.5 km to 11.5 km) and during flight B in seven layers (8.0 to 11.5 km) (Table 2). The vertically averaged size-shape distributions for flights A and B are shown in Fig. 2. The habit distribution, the maximum ice crystal size and fractional contribution of small ice crystals are rather different for the two cases. Small ice crystals with $D_{max} < 100$ µm contribute as much as 79
240 % to the total projected area and optical thickness for flight A, as compared with 27 % for flight B. Considering the large ice crystals only, during flight A the largest contributions to the projected area come from column aggregates (30 %), bullet rosettes (29 %), whereas during flight B, from column aggregates (46 %) and plate aggregates (31 %). Comparing Fig. 2 against Fig. 10 in Jackson et al. (2015) establishes the degree to which the data from these two flights were representative of
245 those observed during other SPARTICUS flights: flight A tends to have lower N(D) than the average observed during other flights whereas flight B tends to have larger N(D) than the observed averages. Overall, flights A and B well represent the range of conditions observed during SPARTICUS.

To investigate the impact of ice crystal sizes on disk and circumsolar radiances, sensitivity tests were also conducted with a lognormal size distribution,

$$250 \quad n(D) = \frac{1}{\sqrt{2\pi}\ln\sigma_{\mathrm{g}}d} \exp\left[-\frac{(\ln D - \ln D_0)^2}{2\ln^2\sigma_{\mathrm{g}}}\right]. \tag{2}$$

Here $\sigma_g$ is the geometric standard deviation (fixed at $\sigma_g = 1.5$) and $D_0$ is the median diameter, for which values of 50, 100, 200, 400, and 800 $\mu$m are considered. The lognormal size distribution covers particles with maximum diameter from 2 to 10 000 µm. The treatment of ice crystal shapes in the tests with a lognormal size distribution is discussed in Sect. 3.4, in connection to Eq. (3).

## 255  3.3  Ensemble-averaged ice crystal optical properties

To obtain the ensemble-averaged optical properties of the ice clouds during flights A and B, the in-situ-measured size-shape distributions are combined with single-scattering properties of individual ice crystals obtained from the database of Yang et al. (2013). In this database, the optical properties are given as a function of wavelength and size ($D_{max}$), shape and roughness of the particle. The
260 three roughness options are: completely smooth (i.e, homogeneous) (CS), moderately rough (MR) and severely rough (SR). The effect of roughness is simulated by randomly distorting the surface slope for each incident ray, assuming a normal distribution of local slope variations with a standard deviation of 0.03 and 0.50 for the MR and SR cases, respectively (Eq. 1. in Yang et al. (2013)). In fact, this treatment does not represent any specific roughness characteristics but attempts instead to

mimic the effects due to non-ideal crystal characteristics in general (roughness effects, irregularities and inhomogeneities like air bubbles).

For each ice crystal size and shape, the cross-sectional area, $A$, the extinction efficiency, $Q_{ext}$, the single-scattering albedo, $\omega$, and the phase function $P_{11}(\gamma)$ at $\lambda$=0.670 μm are obtained from the database, using the closest $D_{max}$ available in the database. The phase function with 498 scattering

angles ($\gamma$ between 0 and 180°) is interpolated to 2011 scattering angles to obtain sufficient angular resolution in the near-forward directions. For single-habit distributions, the in-situ-measured size distribution $N(D_{max} > 100\,\mu m)$ of either flight A or B is combined with the optical properties of that habit and then integrated over the size distribution to obtain the vertical profiles of the ensemble averaged optical properties $K_{ext}$, $\omega$, and $P_{11}(\gamma)$.

For the IC-PCA based habit distributions, the optical properties of each habit are weighted by the IC-PCA fractions before size integration. Hereafter, the optical properties based on the in-situ measured IC-PCA size-shape distributions of flight A and B are referred to as $large_A$ and $large_B$, respectively, since small ice crystals were not classified by shape. Finally, when studying the sensitivity of disk and circumsolar radiances to the concentration of small ice crystals, $large_A$ and $large_B$

are combined with the optical properties of the four alternative in situ based size distributions of small hollow column crystals (see Sect. 3.2).

In the radiative transfer simulations, however, the cloud optical thickness integrated from the in situ based size-shape distributions ($\tau_c = \int K_{ext}(z)dz$, where $z$ is altitude) is not used. Instead, the same user-specified $\tau_c$ for each size-shape distribution in the sensitivity tests is used. This overcomes

the effects related to different area-ratios of the crystal habits and to enable the comparisons of the size-shape distributions of flights A and B. By fixing the cloud optical thickness, the in situ concentrations of the size-shape distributions are adjusted by a uniform factor across all shapes and sizes. Furthermore, when comparing the modeled radiances with those measured with the SAM instrument, $\tau_c$ is adjusted so that modeled radiances in the disk region agree closely (i.e., within

$\approx$ 3 %) with the measurements. This often leads to values of $\tau_c$ that deviate from those retrieved from the SAM ($\tau_{SAM}$) during flights A and B. The values of $\tau_{SAM}$ vary from 0.1 to 2.1 during flight A and from 0.3 to 3.6 during flight B (Fig. 3), indicating that the clouds were not horizontally homogeneous during the flights. This further justifies the approach of using a fixed cloud optical depth because variations in $\tau_{SAM}$ over the course of a flight show that exact agreement between

retrieved and in-situ-based optical depth should not be expected.

### 3.4 Ice cloud phase functions

Ice crystal phase functions play a key role in determining the angular distribution of disk and circumsolar radiances. Therefore, to aid the interpretation of the radiance comparisons, the impact of ice crystal size, habit and roughness on $P_{11}$ (integrated over the cloud depth and the size-shape dis-

tribution) is considered in Figs 4 and 5. First, the impact of ice crystal size on $P_{11}$ is demonstrated in

Fig. 4a, which shows phase functions for three lognormal size distributions with $D_0 = 50$, 200 and 800 μm and for the in-situ based $large_A$ and $large_B$ distributions, assuming SR ice crystals. The phase functions of the lognormal size distributions defined by Eq.(2) are calculated as a weighted sum over the six habits considered in Table 1:

$$P_{11}(\gamma, D_0, \sigma_g) = \sum_{i=1}^{6} w_i P_{11}^i(\gamma, D_0, \sigma_g), \tag{3}$$

where $P_{11}^i(\gamma, D_0, \sigma_g)$ is the phase function corresponding to the lognormal size distribution for habit $i$ and $w_i$ is the weight factor. Here, the weight factors $w_i$ were chosen to equal the fractional contributions of each habit to the projected area for the $large_A$ distribution (see Sect. 3.2). This treatment ensures that independent of the value of $D_0$, the fractional contributions by different habits remain the same, which helps to isolate the effect of crystal size. Tests were also conducted with $w_i$ based on the $large_B$ distribution, and similar effects of crystal size were found (not shown).

By comparing the $P_{11}$ for the different lognormal distributions in Fig. 4a it is seen that ice crystal size has systematic effects on the phase function. With increasing $D_0$, the diffraction peak becomes sharper and narrower, so that the phase function increases at very-near forward directions but decreases at larger scattering angles up to a few degrees. The phase functions computed for the in-situ based $large_A$ and $large_B$ distributions follow this same pattern. Due to to the presence of larger ice crystals during flight B than during flight A (Fig. 2), the angular slope of $P_{11}$ is steeper for flight B than for flight A. The values of $P_{11}$ decrease by roughly four orders of magnitude from the exact forward-scattering direction $\gamma = 0°$ to $\gamma = 10°$ for flight A and by nearly five orders of magnitude for flight B.

The impact of ice crystal habit on $P_{11}$ is illustrated in Fig. 4b–c, which shows the phase function differences between $P_{11}$ of single-habit distributions and the $large_A$ and $large_B$ distributions, respectively. The differences in $P_{11}$ related to ice crystal habit are relatively subtle compared to the large angular slope of $P_{11}$ in near-forward directions, but not negligible. At scattering angles of 0 to 0.1°, plates yield the strongest forward scattering (over 35 % stronger than that of the observed $large_A$ or $large_B$ habit distributions) and bullet rosettes or plate aggregates the weakest scattering (up to 25 % weaker than that of the $large_A$ or $large_B$ distributions). Furthermore, while the $P_{11}$ of plates is lower than that of most other SR habits at angles of 0.3–1°, it is highest among the habits considered at angles of 2–10°. At these angles, plates yield up to 60 % and 80 % larger $P_{11}$ than the observed $large_A$ and $large_B$ distributions, respectively, while columns and column aggregates yield ≈20 % lower values. The impact of habit depends somewhat on the assumed ice crystal roughness; in particular, for CS crystals, $P_{11}$ of plates exceeds that of the $large_A$ and $large_B$ distributions by over 80 % in the very near-forward directions of 0–0.1°.

Figure 5a compares the $P_{11}$ corresponding to the three roughness assumptions for the $large_A$ size-shape distribution, while Figs. 5b–c show the relative differences between the SR and MR ice crystals and the completely smooth (CS) ice crystals for the $large_A$ and $large_B$ distributions. The

$P_{11}$ for rough ice crystals is lower than that for CS crystals in very-near-forward scattering directions, but larger at larger angles, starting from $\approx 0.8°$ for MR crystals and from $\approx 1.7°$ for SR crystals. Furthermore, the $P_{11}$ of MR crystals exceeds that for SR crystals up to $\approx 6°$ but at larger angles, SR crystals yield the largest $P_{11}$. Quantitatively, the impact of roughness is very large and clearly exceeds that of ice crystal habit. The relative difference between MR and CS crystals peaks at 4–5°, reaching 400 % for $large_A$ and over 700 % for $large_B$, while the difference between SR and CS crystals is largest at 7–8°(up to 500 % for $large_A$ and over 600 % for $large_B$).

The phase function differences seen in Fig. 5 are mainly related to rays that are transmitted through an ice crystal, entering and exiting through parallel crystal faces. If the crystal faces are exactly parallel, the phase function contribution by this process is concentrated at very small scattering angles (if finite-size effects are accounted for, as in the Yang et al. (2013) database) or even in the exact forward direction (i.e., delta-transmission), in the limit of geometric optics. However, in the case of MR and SR crystals, the ice crystal surface slopes are distorted randomly for each incident ray, which, in effect, eliminates ray paths that pass through exactly parallel faces. Hence for both MR and SR crystals, $P_{11}$ is lower than that for CS crystals in very-near-forward scattering directions, but larger at larger angles (see Fig. 5). Furthermore, almost the same amount of scattered energy is removed from the very near-forward directions (up to  0.5°) for MR and SR crystals, and added at larger scattering angles, however, with a different angular distribution. The standard deviation of local slope variations assumed in the case of MR crystals is $\sigma = 0.03$, implying that the scattering angle is typically modified by a few degrees, whereas for SR crystals with $\sigma = 0.50$, the scattered energy is distributed over a much larger range of scattering angles. This explains why the relative difference between MR and CS crystals in Fig. 5 peaks at smaller scattering angles than the difference between SR and CS crystals, and why $P_{11}$ for MR crystals exceeds that for SR crystals up to 6°.

## 4   Disk and circumsolar radiances: sensitivity tests

In the sensitivity simulations, the size-shape distribution and roughness of ice crystals, ice cloud optical thickness $\tau_c$ and aerosol optical thickness $\tau_a$ are varied. When not otherwise stated the following parameter settings are used: (1) either the $large_A$ or $large_B$ size-shape distribution of large severely rough ice crystals, with no small crystals with $D_{max} < 100\,\mu m$; (2) cloud optical thickness $\tau_c = 1.6$; (3) atmospheric and aerosol properties corresponding to flight A; (4) aerosol optical thickness $\tau_a = 0.09$; (5) solar zenith angle of $\theta = 40°$. The simulated radiances (in $\mathrm{Wcm^{-2}\mu m^{-1}sr^{-1}}$) are shown as a function of the angular distance from the center of the sun ($0°$) out to $8°$ when looking towards the Sun from the ground (see Fig. 1).

### 4.1 Sensitivity of radiances to optical path

To demonstrate the impact of aerosol and cloud optical thicknesses on radiance, Fig. 6 shows the simulated radiances for an aerosol and cloud-free atmosphere (i.e., "gases only") and for cloud-free (with gases and aerosols) and cloudy (gases, aerosol and ice cloud) atmospheres. The $large_A$ size-shape distribution is used for the cloud, and two values are considered both for cloud ($\tau_c$=0.2 and $\tau_c$=1.6) and aerosol ($\tau_a$=0.09 and 0.166) optical thickness. From Fig. 6 it is seen that in the "gases only" case there is a huge contrast between the very strong radiances in the disk area (1000–2400 $\mathrm{Wcm^{-2}\mu m^{-1}sr^{-1}}$) and the weak and almost constant radiances ($\approx$0.001 $\mathrm{Wcm^{-2}\mu m^{-1}sr^{-1}}$) in the circumsolar region. In the presence of aerosols with $\tau_a$=0.09 or $\tau_a$=0.166, the disk radiances are 11 % and 18 % smaller and the circumsolar radiances are one to two orders of magnitude greater than in the "gases only" simulation. While the circumsolar radiances are $\approx$ 60 % larger for $\tau_a = 0.166$ than for $\tau_a = 0.09$, the relative difference between these cases decreases to less than 20 % when an ice cloud is included, even for $\tau_c = 0.2$.

In the presence of a cirrus cloud, the circumsolar radiances are orders of magnitude greater than in the "gases only" and cloud-free cases as seen from Fig. 6. The most striking effects, both in the absolute values and in the angular dependence, are seen in the angular region between the limb of the solar disk and 1°, where in the cloudy cases the radiances are between 100 and 0.8 $\mathrm{Wcm^{-2}\mu m^{-1}sr^{-1}}$ as compared with $\sim 0.1$ $\mathrm{Wcm^{-2}\mu m^{-1}sr^{-1}}$ for the cloud-free cases and $\sim 0.001$ $\mathrm{Wcm^{-2}\mu m^{-1}sr^{-1}}$ for the gases only case. The increase in diffuse radiance in the presence of a cirrus cloud is due to the strong forward-scattering peak of ice crystals, whereas the smaller disk radiances are due to the larger total optical thickness. The disk radiance decreases monotonically with increasing $\tau_c$, being 74 % less for $\tau_c = 1.6$ than $\tau_c = 0.2$. This is due to the decrease in direct solar radiation; the diffuse radiation in the disk region is, in fact, larger for $\tau_c = 1.6$ than $\tau_c = 0.2$ (see the insert in Fig. 7). In contrast, the circumsolar radiance is on average 140–170 % larger for $\tau_c = 1.6$ than $\tau_c = 0.2$, depending on the assumed $\tau_a$. However, as demonstrated in Fig. 7, the increase of diffuse radiance with $\tau_c$ is not linear, and when attenuation becomes strong enough, the amount of diffuse radiation decreases with increasing $\tau_c$, both in the disk and circumsolar regions.

### 4.2 Sensitivity of the radiances to ice crystal sizes

The impact of ice crystal size on disk and circumsolar radiances is illustrated in Fig. 8, for lognormal size distributions with habit weight factors based on the IC-PCA habit distribution of flight A (see the first paragraph of Sect.3.4). As expected based on previous research (e.g. DeVore et al., 2009; Segal-Rosenheimer et al., 2013) and the phase functions in Fig. 4, the ice crystal size has systematic effects on radiances in the vicinity of sun. With an increasing median diameter $D_0$, the radiances increase at very small angles (up to $\approx 0.3 - 0.4°$ from the center of the sun) but decrease at somewhat larger angles, with largest effects at $\approx 0.5 - 2°$ for CS crystals and $\approx 0.5 - 1°$ for SR crystals. At angles

of several degrees, the impact of $D_0$ becomes small especially for SR crystals. Decreasing $D_0$ has opposite effects. For example, doubling $D_0$ from 200 to 400 μm decreases the radiance at 0.5–1° by up to 45 %, while halving $D_0$ from 200 to 100 μm increases it by up to 80 %, for the optical thickness $\tau_c = 1.6$ considered here. The effects in the solar disk area are somewhat smaller, $\approx 15$ % for SR crystals and $\approx 25$ % for CS crystals. These difference arise, to a large part, from the effect of

ice crystal size on the diffraction peak (see Fig. 4).

A related issue is the effect of small ice crystals, for which the measurements are quite uncertain. To probe the impact of uncertainties in the measurements of small ice crystals, the effects of their concentration on the disk and circumsolar radiances are simulated. Simulations are made with the $large_A$ and $large_B$ distributions together with 0–200 % of the measured concentration of small

column-shaped ice crystals ($small_{0\%}$, $small_{50\%}$, $small_{100\%}$, and $small_{200\%}$). In these simulations, ice crystals are severely rough and $\tau_c = 1.6$. The angular dependence of the total radiances simulated with 0 and 100 % of the measured small-crystal concentrations are shown in Fig. 9a at angles of 0 to 4° from the center of the sun. Regardless of the small crystal concentration, the radiances at angles larger than 5° are within 3 % of each other during flights A and B. Because the same cloud optical

thickness is assumed for all the size distributions, including small ice crystals necessarily decreases the concentration of large ice crystals. This acts to decrease the near-forward radiances in the disk region and just around it and to increase the circumsolar radiances at angles between 0.5 and 5° from the center of the sun. Again, this is due to the wider forward-scattering peak of the small ice crystals.

Quantitatively, the impact of the assumed concentration of small ice crystals is substantial and

425 somewhat larger for flight A than flight B (see Fig. 9b–c). Compared to the cases with large ice crystals only, the relative reduction in radiance due to small ice crystals is largest near the edge of the solar disk, amounting up to $-47$ % for flight A and $-22$ % for flight B. The largest relative increases occur at 1–2° from the center of the sun. For flight A, the maximum differences to the case with large ice crystals only are 95 %, 111 % and 123 %, and for flight B, 33 %, 55 % and 84 %, when

assuming 50 %, 100 % and 200 % of the observed concentration of small ice crystals, respectively. The impacts of size on diffuse radiation tend to be opposite in the disk and circumsolar regions and partly cancel each other leading to smaller differences when averaged from 0° to 3°. Even so, for a fixed cloud optical thickness of 1.6, including 100 % of the observed concentration of small crystals, the average total radiance in this angular range is 26 % larger than in the case with large crystals only

for flight A, and 11 % larger for flight B. The amount of diffuse radiation in this angular range is 44 % and 53 % of the total radiance for $large_A$ and $large_B$, respectively. The corresponding fractions in the disk region are 41 % and 51 %.

## 4.3 Sensitivity of the radiances to shape and roughness of ice crystals

Sensitivities of the disk and circumsolar radiances to the size-shape distributions of large ice crys-

440 tals are addressed by comparing results for the six single-habit distributions and the measured habit

distributions of flights A and B. The radiances simulated with $large_A$ and $large_B$ size-shape distributions are compared in Fig. 10a. For $large_B$, the total radiance in the disk region is 10 to 20 % larger than for $large_A$, and the circumsolar radiance is smaller by up to 30 %, even though the same $\tau_c$ is assumed in both cases. This occurs because the ice crystal population for flight B results in a stronger and narrower forward-scattering peak in $P_{11}$ as noted already from Fig. 4. As the optical thickness is the same in both cases, the differences in the total radiances arise from differences in the diffuse component.

The relative differences between the six single-habit distributions and the $large_A$ or $large_B$ distributions are shown in Fig. 10b–c for total radiances. The differences in radiances follow the differences in $P_{11}$ of the habit distributions shown in Fig. 4. In the disk region the difference between different habit distributions reaches at most 15 %. The impact of habit, however, differs between flights and therefore depends on the size distribution. Based on the circumsolar radiances, the habits can be divided into two groups; column-like (column, column agg and bullet rosette) and plate-like crystals (plate, plate agg and irregular). Column-like crystals tend to result in a steeper angular slope in radiances, producing larger diffuse radiances in the disk region and smaller radiances in the circumsolar region than plate-like crystals do. In the circumsolar region, plate and column agg tend to differ most from each other regardless of the size distribution. The relative differences in the circumsolar region between the single-habit distributions and CPI-based habit distributions reach up to 60 % for flight A and up to 80 % for flight B, similarly to the phase functions differences in Fig. 4. The impact of ice crystal habit also depends on the cloud optical thickness. Generally, as $\tau_c$ increases and multiple scattering becomes more important, the relative differences in diffuse radiances between different habits are reduced.

The impact of ice crystal roughness on the radiances is depicted in Fig. 11. Consistent with the large phase function differences in Fig. 5, the impacts of roughness on the radiances are substantial: rough crystals yield smaller diffuse radiances than smooth crystals at angles smaller than 1 to 2.5° but larger diffuse radiances at angles larger than that. In the disk region, SR and MR crystals produce almost identical radiances, which are within 1 % of each other, but 15 % to 21 % below those of smooth crystals, depending on the flight. In the circumsolar region at angles smaller than 7°, MR crystals produce larger radiances than the SR crystals, the relative differences being largest at angles of 2° to 3°, up to 140 % for flight A and 195 % for flight B. The relative differences between MR and CS crystals are largest at angles of ∼4°, reaching up to 425 % for flight B. Correspondingly, the maximum relative differences between SR and CS crystals occur at angles larger than 6°, reaching 240 % for flight B. These angle-dependent radiance differences between different roughness assumptions follow the $P_{11}$ differences shown in Fig. 5. The relative differences in radiances are, however, not quite as large as those in $P_{11}$, and they decrease somewhat with increasing $\tau_c$ (here, $\tau_c$=1.6), due to the effects of multiple scattering. In any case, roughness has a large impact on both the disk and the

circumsolar radiances, and these differences clearly exceed the corresponding differences between different SR habits (compare Figs 10 and 11)

The roughness and shape of the particles also impact the fractional contribution that diffuse radiation makes to the total radiance in the range of 0 to 3° typically measured by pyrheliometers. For the cases considered here, the fractional contribution of diffuse radiation to total radiation is ≈ 10 percentage points larger for CS than SR or MR ice crystals. The impact of shape distribution is somewhat smaller, between 4 and 7 percentage points.

### 4.4 Summary of sensitivity tests

The results of the sensitivity tests are summarized in Table 3, where the importance of various ice cloud properties on the disk and circumsolar radiances in four angular regions is characterized in a semi-quantitative manner. Trivially, the direct solar radiation depends only on the cloud optical depth $\tau_c$ through Beer's law. The optical depth also impacts strongly the magnitude of the diffuse radiance but not so much its angular distribution in the circumsolar region (see Fig. 7). Ice crystal size has a large impact on circumsolar radiance at angles close to the sun, due to the well-known impact of particle size on the diffraction peak. Assumptions about ice crystal roughness influence the circumsolar radiation very strongly especially at angles larger than about 2–3°. In comparison to the effects of ice crystal size and roughness, the impact of ice crystal habit is moderate, being most pronounced in the outermost region (3–8°) considered in Table 3.

The effects of particle roughness, size and habit on the distribution of diffuse radiation are closely linked to the corresponding effects on phase function. The impact of particle size on $P_{11}$ is well known, and it has been previously discussed in the context of circumsolar radiation (DeVore et al., 2009, 2012; Segal-Rosenheimer et al., 2013; Reinhardt et al., 2014). The impacts on near-forward scattered radiation due to ice crystal roughness and habit have also been considered (DeVore et al., 2012; Segal-Rosenheimer et al., 2013; Reinhardt et al., 2014). However, the present study extends the knowledge gained in previous research (DeVore et al., 2009, 2012; Segal-Rosenheimer et al., 2013; Wilbert et al., 2013; Reinhardt, 2013; Reinhardt et al., 2014) as we simulate in detail the angular distribution of phase function and radiances, instead of circumsolar irradiances integrated over some angular range (as in Reinhardt et al. (2014)).

The potential applications of these results include remote sensing of optical thickness, ice cloud properties based on the radiance field near the sun, and the design of concentrating solar energy applications. For example, when retrieving cloud optical thickness from measurements of direct solar radiation from ground using instruments with a typical half-opening angle of 3°, one needs to account for the fact that there can be a substantial amount of diffuse radiation within the instrument's field of view (Mauno et al., 2011; Segal-Rosenheimer et al., 2013; Reinhardt et al., 2014). Similarly, when evaluating the potential of concentrating solar collectors with typical half-opening angles of ≈ 1 °(Bui and Monger, 2004; Wilbert et al., 2012, 2013), one has to consider the narrower field

of view as compared with measurements of "direct" solar radiation. Our sensitivity tests reveal that in the presence of an ice cloud, the diffuse radiation in the angular range from 0 to 3° is especially sensitive to the roughness and size distribution of the ice crystals, with the shape of ice crystals being less important.

Overall, the sensitivity tests highlight the important role of ice crystal size distribution and roughness on the distribution of radiances in the vicinity of the sun. Yet, in-situ microphysical measurements yield no information on roughness and only very uncertain information on small ice crystals. This motivates the study of how assumptions related to these factors impact the comparison between simulated and measured radiances in Sect. 5.

## 5  Comparison of the simulated and measured radiances

During SPARTICUS, disk and circumsolar radiances were measured with the SAM instrument of Visidyne Inc. located at the SGP site. For both flights A and B, horizontal cross sections of SAM measurements from three different times are selected for comparison. Note that the radiance arriving at different sensors comes from different parts of the cloud. To assure that the observed angular dependence of radiance is not due to cloud inhomogeneity, only cases where the radiance distributions measured to the right and left of the sun (the "hp" and "hn" curves; see Fig. 1) are similar are considered. The cloud scenes for the times selected for comparing SAM measurements and simulated radiances are shown in Fig. 12. The goal is to reproduce these radiances using the in-situ-based size-shape distributions of ice crystals. The simulations are conducted both with and without the contribution of small ice crystals, assuming 100 % of the measured small-crystal concentration in the former case. The atmospheric and aerosol properties of flights A and B are used in the simulations. Due to the large scale inhomogeneity of the clouds, the cloud optical thickness $\tau_c$ was adjusted separately for each case, based on the criterion that the simulated radiance averaged over the solar disk should be within 3 % of the SAM measurements. The resulting values of $\tau_c$ are listed in Tables 4 and 5 for flights A and B respectively, along with $\theta$ of the selected SAM measurement times, and the total apparent optical thickness (cloud + aerosols) retrieved from SAM assuming that the disk radiance consists of direct solar radiation only. The derived values of $\tau_c$ depend not only on the measurement time but also on the assumptions about ice crystal roughness and small ice crystals. In particular, for a given optical thickness, stronger disk radiances are produced by smooth compared to rough crystals (see Fig. 11), and consequently, larger $\tau_c$ is needed to match the SAM measurements in the case of smooth compared to the case of rough ice crystals. Further, $\tau_c$ tends to be larger than that reported by the SAM. This is in line with DeVore et al. (2012) who found that the $\tau_{SAM}$ needs to be corrected upward to account for forward scattering of ice crystals.

The simulated radiances are compared with the selected SAM measurements in Figs. 13 and Fig.14 for flights A and B, respectively. Both the simulated and the SAM-measured radiances shown

are horizontal profiles (to the left and right, see Fig. 1) from the center of the sun out to $8°$. As the simulated clouds are horizontally homogeneous, the profiles to the left and right from the center of the sun are averaged, whereas for SAM, they are shown separately as the "hp" and "hn" curves. As exhibited in these images, when the aureole intensity drops below the sensitivity limit of the SAM 300 solar disk imager, a gap results starting somewhere at or beyond the disk edge, e.g., $\approx 0.27°$, and extending out to $\approx 0.6°$, where the solar aureole imager begins its measurements. When both the forward scattering and the optical depth are sufficiently large, the aureole profile can be within the sensitivity range of the disk imager and the gap disappears.

First, it is noted that while (by construction) the average simulated disk radiances agree closely with the SAM measurements, the angular slope measured is not quite consistent with the limb-darkening profile used in the simulations. The reasons for this discrepancy are not clear and should be scrutinized in future work. Second, considering the circumsolar radiances, the simulations with SR ice crystals better capture the measured angular dependence than do the simulations for CS crystals. The use of CS ice crystals overestimates near-disk radiances and underestimates the radiances at angles larger than about $3°$. It is further noted from Figs 13 and 14 that excluding small crystals decreases the radiances at angles smaller than $3°$ and by doing so, tends to improve the comparison of radiances at these angles. Overall, it appears that the angular dependence produced by large SR crystals is most similar to the measurements, even though it tends to overestimate the radiances at angles larger than $\approx 6°$ in most cases.

The systematically better performance of SR than CS ice crystals in simulating the measured radiances in the circumsolar region suggests that the SR crystals better approximate the phase function of ice crystals present during flights A and B, at least in near-forward directions. Furthermore, the SR crystals are more consistent with the measured radiances than the MR crystals. The use of MR crystals results in radiances that exceed those for CS and SR crystals and also the measurements between angles of $\approx 1°$ and $6°$, even when small ice crystals are not accounted for. Referring to the discussion in Sect. 3.4, the better performance of SR compared to CS crystals suggests that ray paths passing through smooth, exactly parallel ice crystal faces are less common in nature than they would be for idealized ice crystals. However, there is no reason to expect that the somewhat ad-hoc approach employed in the Yang et al. (2013) database to represent ice crystal "roughness" (or rather, non-ideal features like roughness, irregularities and inhomogeneity in general) would result in a perfect description of $P_{11}$. Even so, these results add to the growing body of evidence (Cole et al., 2014; Ulanowski et al., 2014) suggesting that the scattering by natural ice crystals most often differs from their idealized counterparts, also in the near-forward directions (DeVore et al., 2012).

## 6 Conclusions

In this study, the amount of diffuse radiance in the solar disk region and in the circumsolar region up to angles of 8° from the center of sun were quantified using a modified version of the Monte Carlo radiative transfer model MC-UniK. The input data for the model were derived from the measured size-shape distributions of two ice cloud cases observed over the ARM's Southern Great Plains measurement site during the 2010 SPARTICUS campaign. This work extends and supports the previous studies on the impact of ice crystals' properties on near-forward scattering and circumsolar radiation (Reinhardt et al., 2014; Segal-Rosenheimer et al., 2013) by modelling radiances instead of irradiances and by conducting systematic sensitivity tests using in situ based size-shape distributions of ice crystals.

In the sensitivity tests, it was found that the disk and circumsolar radiances depend substantially on the ice crystal properties (roughness and size-shape distribution) through their impact on the phase function, in line with previous research (Reinhardt et al., 2014; Segal-Rosenheimer et al., 2013; DeVore et al., 2012). Specifically:

 – Of all parameters considered, assumptions about ice crystal roughness (or non-ideal features in general) were found to be most important. The use of moderately or severely rough ice crystals instead of completely smooth crystals leads to reduced radiances in the solar disk region while substantially increasing radiances in the circumsolar region at angles larger than ≈ 1–2.5°, with maximum differences as large as 400 % between MR and CS crystals and 200 % between SR and CS crystals.

 – Ice crystal size distribution is also important for the angular distribution of circumsolar radiance. With increasing ice crystal size, the diffraction peaks becomes sharper and narrower, so that disk radiances increase but radiances at angles of ≈0.5°–5° decrease. Increasing the portion of small ice crystals has the opposite effect. In particular, if 100 % of the measured but uncertain concentration of small ice crystals is included in the calculations, radiances at ≈1–2° from the center of the sun can be up to ≈100 % larger than in the case with only large ice crystals.

 – Column-like crystals tend to yield radiances with a steeper angular slope than plate-like crystals, as they produce more diffuse radiance in the disk region and less in the circumsolar region than plate-like crystals. The relative differences between all single-habit distributions and the actually measured habit distributions were less than 10 % in the disk region but up to 80 % at angles larger than 4° from the center of the sun.

The quantitative results listed above depend on the cloud optical thickness and solar zenith angle. In general, an increasing path length through the cloud acts to reduce the radiance contrast between the disk region and the circumsolar region, and the impact of the phase function. Changes in aerosol

optical thickness also affect the absolute values of the radiances in the presence of an ice cloud, but not significantly their angular dependence.

Simulated radiances were compared with ground-based measurements with the SAM instrument for three measurement times during both flights A and B. It was found that SR ice crystals mimicked the measured circumsolar radiances better than either the MR crystals (which overestimated the radiances at angles of a few degrees) or the CS crystals (which invariably underestimated the radiances at angles larger than ≈3°). In some cases, the agreement was better when crystals smaller than 100 μm were neglected from the measured size distribution, suggesting that the measurements may have overestimated the concentration of small crystals. These results add to the growing body of evidence suggesting that natural ice crystals tend not to be pristine (Cole et al., 2014; Ulanowski et al., 2014).

Even though we had detailed information about the size-shape distribution of ice crystals of the clouds studied, the observed radiances could not be reproduced perfectly. There are several factors that possibly contribute to this. Part of the discrepancies can be surely attributed to the non-perfect spatiotemporal collocation of the in situ and SAM measurements. It is also quite possible that the simplistic ad hoc scheme employed to mimic the effects of roughness, non-ideality and internal structures on scattering is not entirely realistic or representative of natural ice crystals. Further, the limb darkening parameterization may not be entirely accurate, and some discrepancies might also be due to the aerosol optical properties chosen. Likewise, there may be some remaining inhomogeneities in the clouds that our analysis did not reveal. Finally, it is entirely possible that the clouds sampled had mixtures of ice crystals with varying degrees of deformation, in which case any one crystal roughness model could not be expected to perform perfectly, but a combination of differently deformed crystals should be used.

In the future, the version of MC-UniK modified for the present work could be used for analyzing a wide range of cirrus cloud and aerosol scenarios and their 3D effects on near-forward radiances. The unique modeling results might be of interest for the design of concentrating solar power systems and for the interpretation of data from instruments intended to measure the direct solar radiation. The results could also be utilized for evaluating the contribution of diffuse solar radiation to the disk radiation in SAM measurements, thereby allowing for a more accurate determination of the "true" direct solar radiation, and hence the optical thickness. Furthermore, they might be exploited for developing methods to retrieve ice cloud properties from measurements of disk and circumsolar radiances; in particular, it might be possible to estimate ice crystal non-ideality from SAM measurements. Finally, the combination of SAM with sun photometer measurements (e.g. AERONET) might allow separating the contributions of large and small particles (e.g., ice crystals vs. aerosols) to optical thickness.

*Author contributions.* TN, GM, PR, MK, AM and PH designed the study. GM provided the airborne in situ data, JD the SAM data and AM the Monte Carlo model MC-UniK. JT was responsible for using the IC-PCA.

MK, TN, PR, AM and PH planned the needed modifications of the MC-UniK, which PH then implemented. PR calculated the atmospheric optical properties. PH combined the in situ data and single-scattering properties, conducted all the MC-UniK simulations and made the figures, except for Fig. 2 which was made by PR. PH, PR, GM and TN wrote the manuscript. All authors discussed the results and commented on the manuscript.

*Acknowledgements.* We wish to thank the two anonymous referees for their time and constructive comments that helped to improve the manuscript. Dennis Villanucci and Andrew LePage are acknowledged for their effort in establishing and maintaining the the SAM instrument deployed at the ARM SGP site during SPARTICUS. Hannakaisa Lindqvist is thanked for providing the IC-PCA model and Robert Jackson for comparing the size distributions measured during SPARTICUS. This work was supported by the Academy of Finland Centre of Excellence (grant no. 272041), the Maj and Tor Nessling Foundation (201600119) and by the Office of Biological and Environmental Research (BER) of the U.S. Department of Energy (DE-SC0008500, DE-SC0014065 and DE-SC0016476 through UCAR subcontract 217-90029). Data were obtained from the Atmospheric Radiation Measurement (ARM) program archive, sponsored by the U.S. DOE, Office of Science, BER, Environmental Sciences Division.

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

**Table 1.** The final habit classes of large ice crystals that are created by combining habit classes of the IC-PCA and further interpreted as Yang et al. (2013) habits. In addition to the IC-PCA based habit distributions, $large_A$ and $large_B$, six single-habit distributions are used to describe the shape of large ice crystals.

| habit class | habits of Yang et al. (2013) | habit classes of IC-PCA |
|---|---|---|
| column | hollow column | columns and bullets |
| column agg | column aggregate with 8 elements | column aggregates and bullet rosette aggregates |
| bullet rosette | bullet rosette | bullet rosettes |
| plate | plate | plate |
| plate agg | plate aggregate with 5 elements | plate aggregate |
| irregular | plate aggregate with 10 elements | irregular |
| large | fractional distribution of habits from in situ data | habits classified using IC-PCA |

**Table 2.** Flight information. $\theta$ is the solar zenith angle during the flights A and B.

| | Flight A | Flight B |
|---|---|---|
| Date | 23 March 2010 | 24 June 2010 |
| Time [UTC] | 16:58-17:56 | 14:35-15:58 |
| $\theta$ [°] | 36.5-42.1 | 42.7-52.3 |
| Cloud altitude [km] | 9.5-11.5 | 8.0-11.5 |
| Model layers with cloud | 8-11 | 5-11 |

**Table 3.** A semi-quantitative summary on the strength of the impacts of microphysical parameters to the direct radiance and four different angular regions of circumsolar radiance. The first row address the impacts due to the ice cloud optical thickness, and the following rows due to size, shape and roughness of ice crystals. In parentheses is given the parameter value against which the relative differences are calculated. The maximum relative strength of the impact is given with symbols: $-$ (impact < 50 %); $+$ (50 % < impact < 100 %); $++$ (100 % < impact< 200 %) and $+++$ ( 200 % < impact) and is based on the conducted sensitivity tests (see Figs 7–11).

| parameter | dir $0.0°$–$0.27°$ | diff $0.0°$–$0.27°$ | diff $0.27°$–$1°$ | diff $1°$–$3°$ | diff $3°$–$8°$ |
|---|---|---|---|---|---|
| optical thickness ($\tau_c$=1.6) | +++ | + | + | + | + |
| median size ($D_0 = 200\ \mu$m) | $-$ | + | +++ | +++ | + |
| small crystals $D < 100\ \mu$m ($large_{A/B}$) | $-$ | + | + | ++ | + |
| shape distribution ($large_{A/B}$) | $-$ | $-$ | $-$ | + | + |
| roughness (CS) | $-$ | $-$ | $-$ | +++ | +++ |

**Table 4.** The values of solar zenith angle $\theta$ and optical thickness of cloud ($\tau_c$), aerosols ($\tau_a$), and gases ($\tau_{gases}$) used in the comparison simulations for flight A. The cloud is described with the size-shape distributions $large_A$ and $large_A + small_{100\%}$ of rough (MR and SR) and completely smooth (CS) ice crystals. The fractional contribution of small ice crystals to cloud optical thickness for the $large_A + small_{100\%}$ size-shape distribution ($f_{small}$) and the total optical thickness (cloud+aerosols) retrieved from the Sun and Aureole measurements (SAM) are also listed.

| $\theta$ [°] | 40.5 | 38.3 | 38.6 |
|---|---|---|---|
| $\tau_{gases}$ | 0.072 | 0.072 | 0.072 |
| $\tau_a$ (AERONET, MFRSR) | 0.09 | 0.09 | 0.09 |
| $f_{small}$, $large_A + small_{100\%}$ | 79 % | 79 % | 79 % |
| $\tau_c$, CS, $large_A + small_{100\%}$ | 0.6 | 1.05 | 2.5 |
| $\tau_c$, MR/SR, $large_A + small_{100\%}$ | 0.6 | 1.0 | 2.4 |
| $\tau_c$, CS, $large_A + small_{0\%}$ | 0.75 | 1.25 | 3.1 |
| $\tau_c$, MR/SR, $large_A + small_{0\%}$ | 0.65 | 1.15 | 2.75 |
| $\tau_{SAM}$ | 0.6 | 1.0 | 2.1 |

**Table 5.** The values of solar zenith angle $\theta$ and optical thickness of cloud ($\tau_c$), aerosol ($\tau_a$) and gases ($\tau_{gases}$) used in the comparison simulations for flight B. The cloud is described with the size-shape distributions $large_B$ and $large_B + small_{100\%}$ of rough (MR and SR) and completely smooth (CS) ice crystals. The fractional contribution of small ice crystals to cloud optical thickness for the $large_B + small_{100\%}$ size-shape distribution ($f_{small}$) and the total optical thickness (cloud+aerosols) retrieved from the Sun and Aureole measurements (SAM) are also listed.

| $\theta$ [°] | 50.4 | 50.0 | 44.3 |
|---|---|---|---|
| $\tau_{gases}$ | 0.074 | 0.074 | 0.074 |
| $\tau_a$ (AERONET, MFRSR) | 0.166 | 0.166 | 0.166 |
| $f_{small}$, $large_B + small_{100\%}$ | 27 % | 27 % | 27 % |
| $\tau_c$, CS, $large_B + small_{100\%}$ | 0.7 | 1.3 | 3.5 |
| $\tau_c$, MR/SR, $large_B + small_{100\%}$ | 0.6 | 1.15 | 3.05 |
| $\tau_c$, CS, $large_B + small_{0\%}$ | 0.75 | 1.45 | 4.0 |
| $\tau_c$, MR/SR, $large_B + small_{0\%}$ | 0.65 | 1.25 | 3.3 |
| $\tau_{SAM}$ | 0.6 | 1.0 | 2.3 |

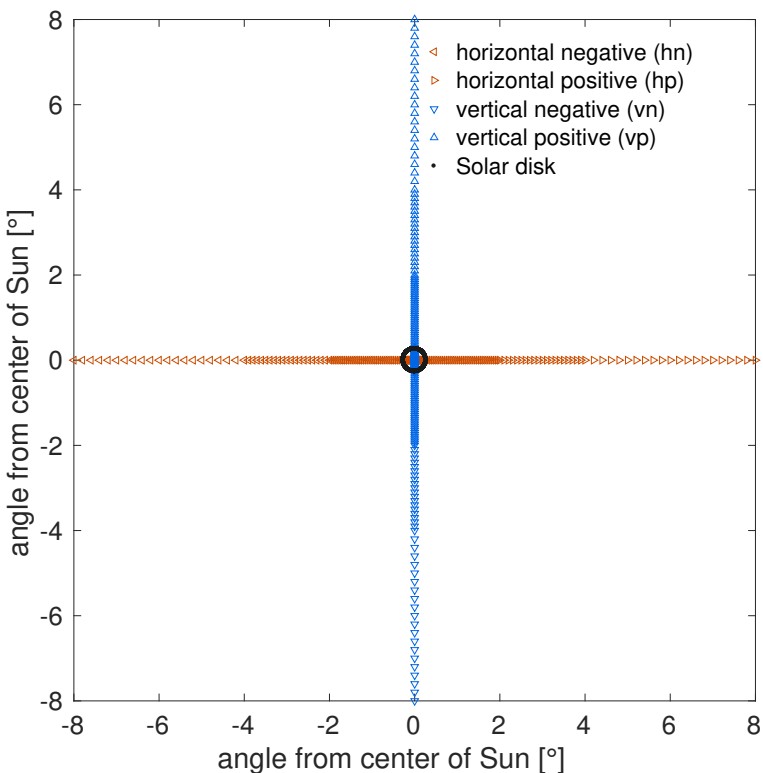

**Figure 1.** Detector positions in the MC-UniK model cover angles from 0 to -8 and 8° from the center of the sun (0°,0°). Both horizontal and vertical cross sections are divided into positive and negative parts (hp; hn; vp; vn). The circle demonstrates the size of the solar disk, with a diameter of 0.534°.

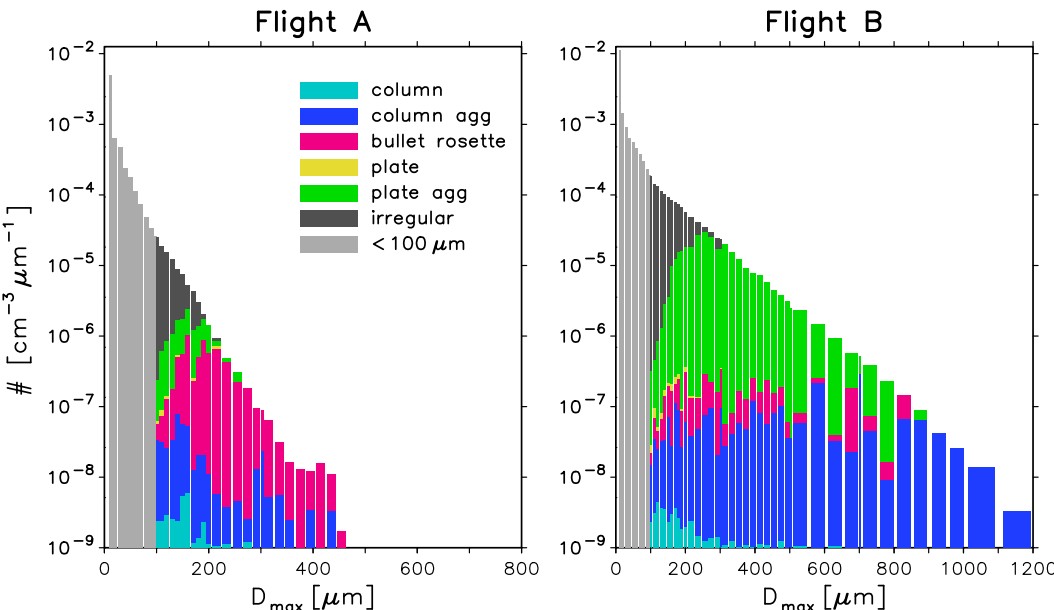

**Figure 2.** Vertically averaged size-shape distribution of in situ measured ice crystals during the flights on 23 March 2010 (flight A) and 24 June 2010 (flight B). These distributions were obtained by weighting fractional habit distributions at each vertical layer by the corresponding particle size distribution. The height of each column indicates the total number of particles in each size range (logarithmic scale on the y-axis). The fraction of particles of each habit is shown with different colors (in a linear scale). Small ice crystals with $D_{\max} < 100\,\mu m$ are shown with gray color. They were treated as columns in the calculations.

.

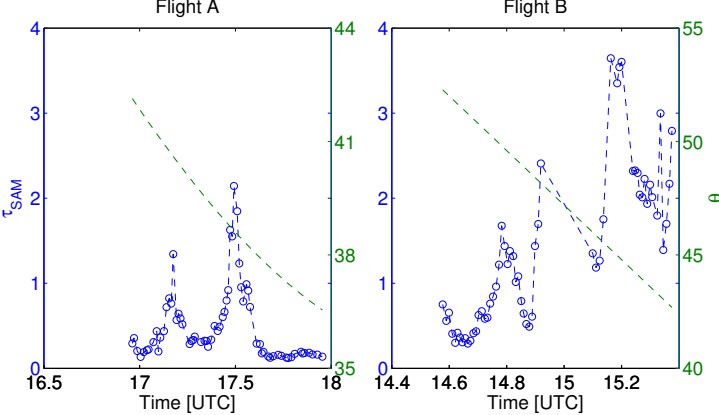

**Figure 3.** Optical thickness and solar zenith angle ($\theta$) as a function of time during the flights A and B derived from the Sun and Aureole measurements at the SGP site.

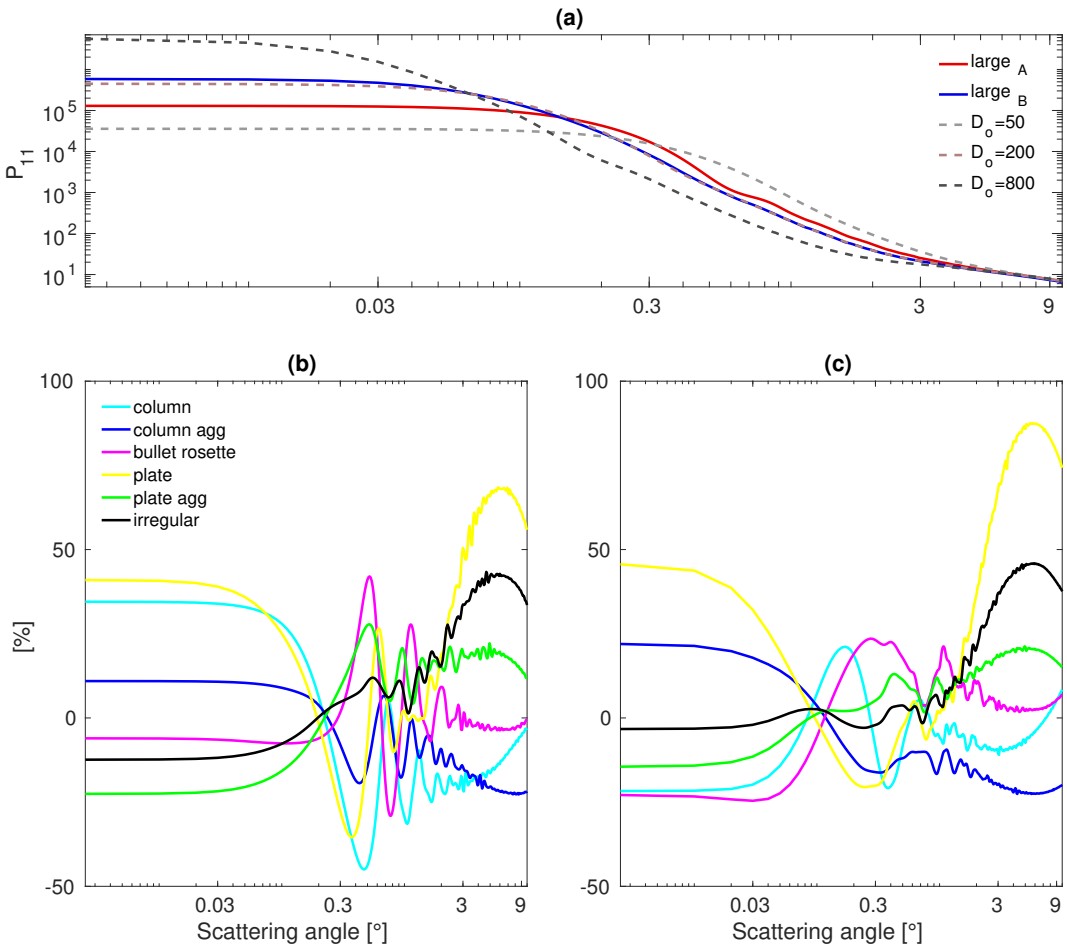

**Figure 4.** Sensitivity of the vertically integrated phase functions to the size-shape distribution of large severely rough ice crystals. (a) The $P_{11}$ of in-situ-based distributions of flights A and B and lognormal distributions with $D_0 = 100/200/400$ $\mu$m . (b) and (c) The relative differences in $P_{11}$ between the six single-habit distributions and the in-situ-based distributions for flight A (left) and B (right).

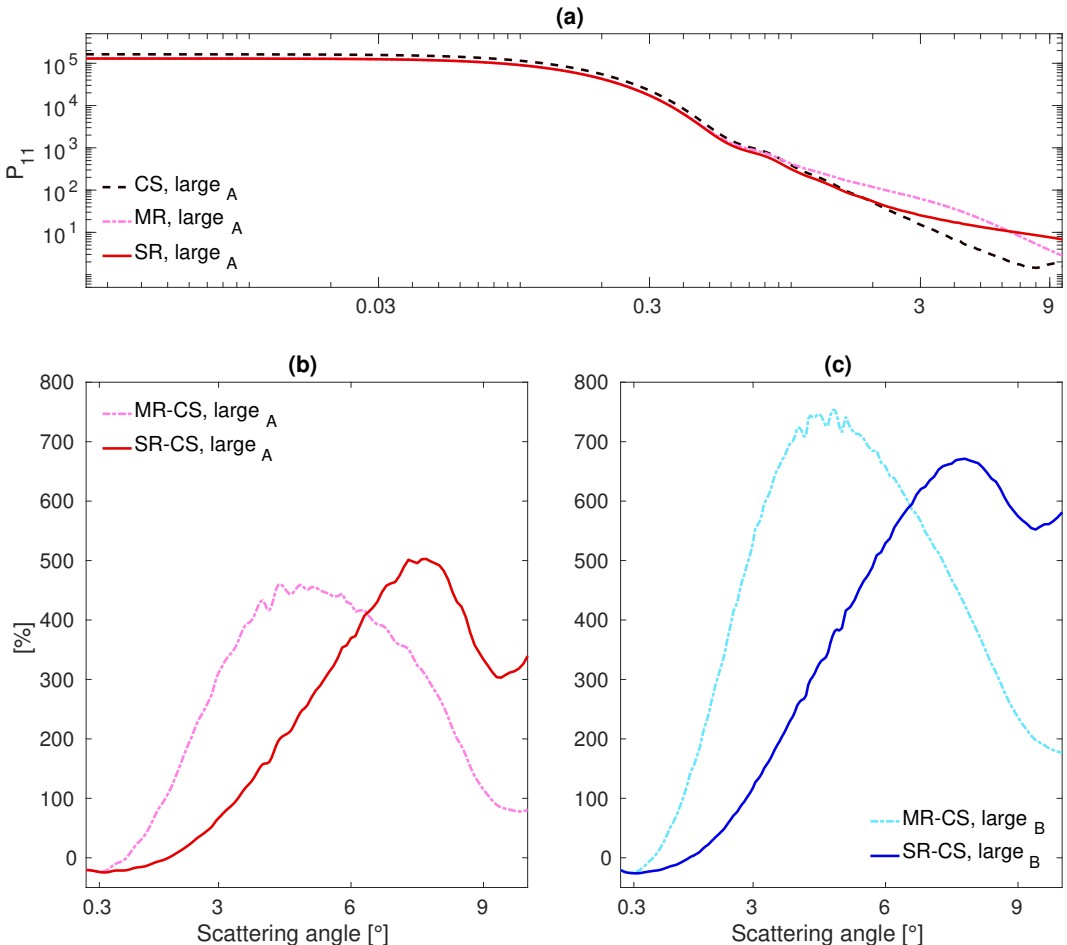

**Figure 5.** Sensitivity of the vertically integrated phase functions to the roughness of large ice crystals. **(a)** The $P_{11}$ of the in-situ-based size-shape distribution of smooth, moderately and severely rough ice crystals of flight A ($large_A$). **(b)** and **(c)** The relative differences in $P_{11}$ between MR and CS ice crystals and between SR and CS ice crystals of flight A and B.

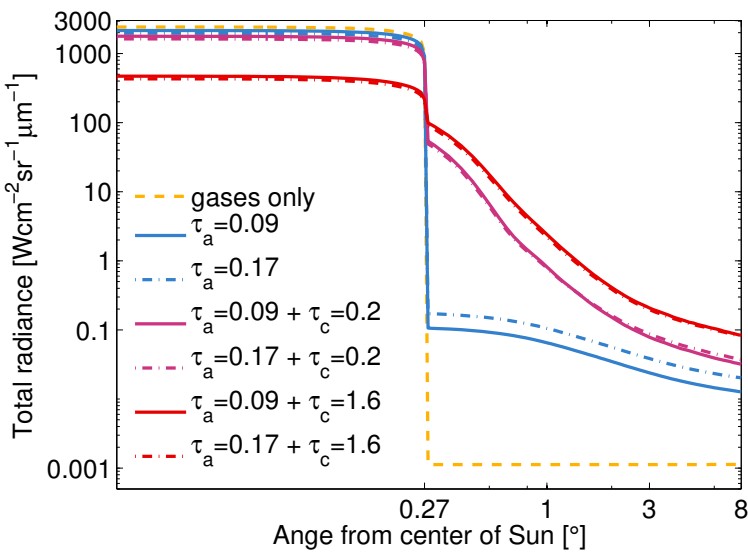

**Figure 6.** Impacts of the aerosol and cloud optical thicknesses on the simulated radiances as a function of angle from the center of the sun out to $8°$. Atmospheric and aerosol properties are based on flight A with either $\tau_a = 0.09$ or $\tau_a = 0.166$. The cloud is described with the $large_A$ distribution of large SR ice crystals using two cloud optical thicknesses, $\tau_c = 0.2$ and 1.6.

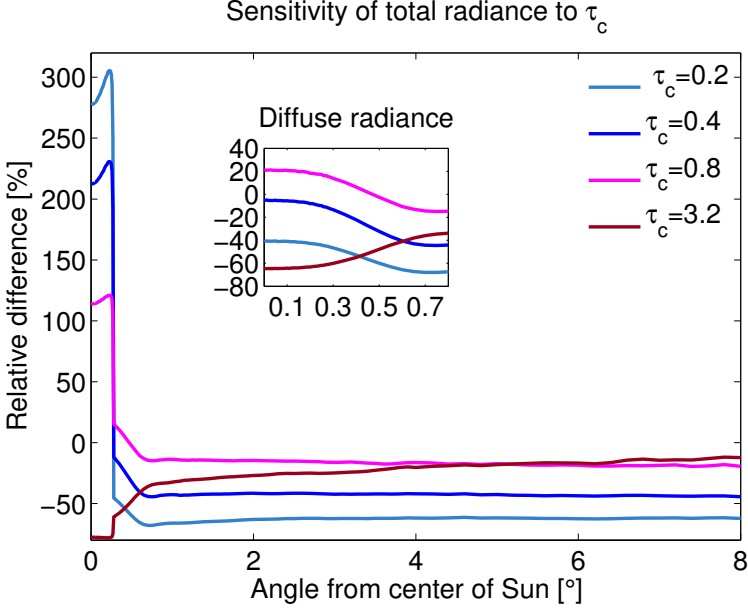

**Figure 7.** Sensitivity of the disk and circumsolar radiances to cloud optical thickness when the cloud is described using the $large_A$ distribution of SR ice crystals. Relative differences between radiances simulated with alternative cloud optical thicknesses ($\tau_c$ of 0.2, 0,4, 0,8, and 3.2) and $\tau_c = 1.6$ are displayed. The insert shows the relative differences in diffuse radiance at angles of $0–0.8°$. In these simulations $\tau_a = 0.09$.

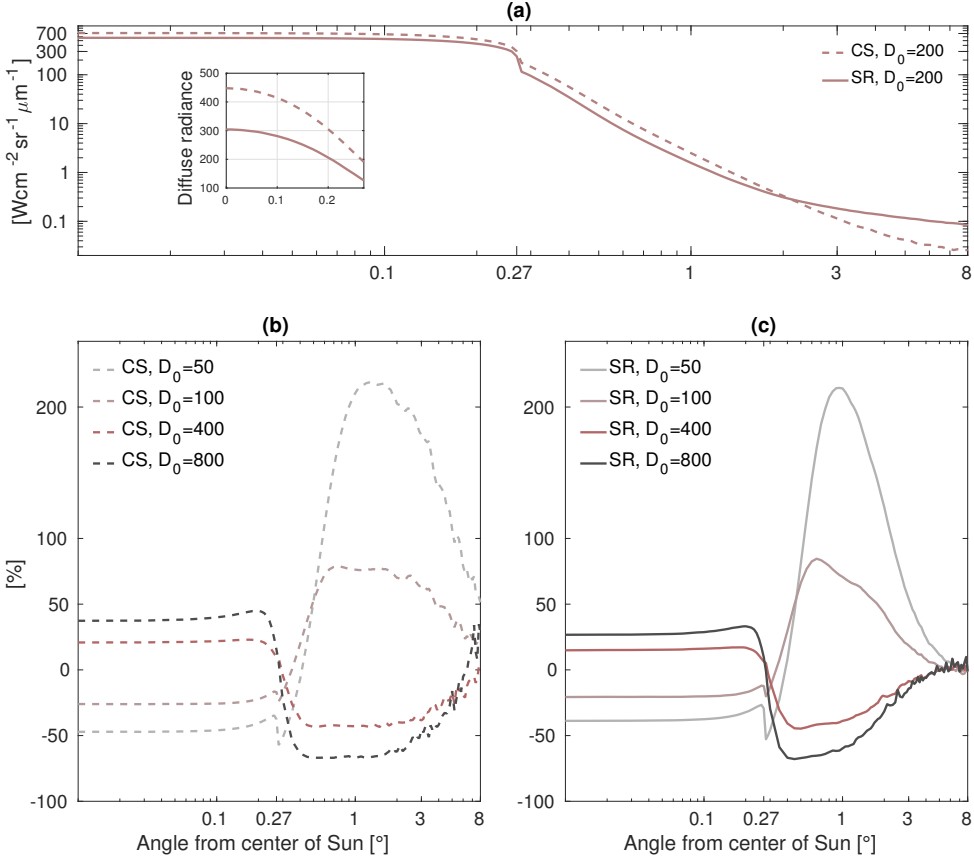

**Figure 8.** Impact of ice crystal size distribution on the disk and circumsolar radiances. **(a)** shows, for reference, the radiances for a lognormal size distribution with a median diameter $D_0 = 200\,\mu m$ for smooth (CS) and severely rough (SR) ice crystals. **(b)** and **(c)** show the relative differences to the case with $D_0 = 200\,\mu m$ for $D_0$=50, 100, 400 and 800 $\mu m$. In these simulations, $\tau_a = 0.09$ and $\tau_c = 1.6$.

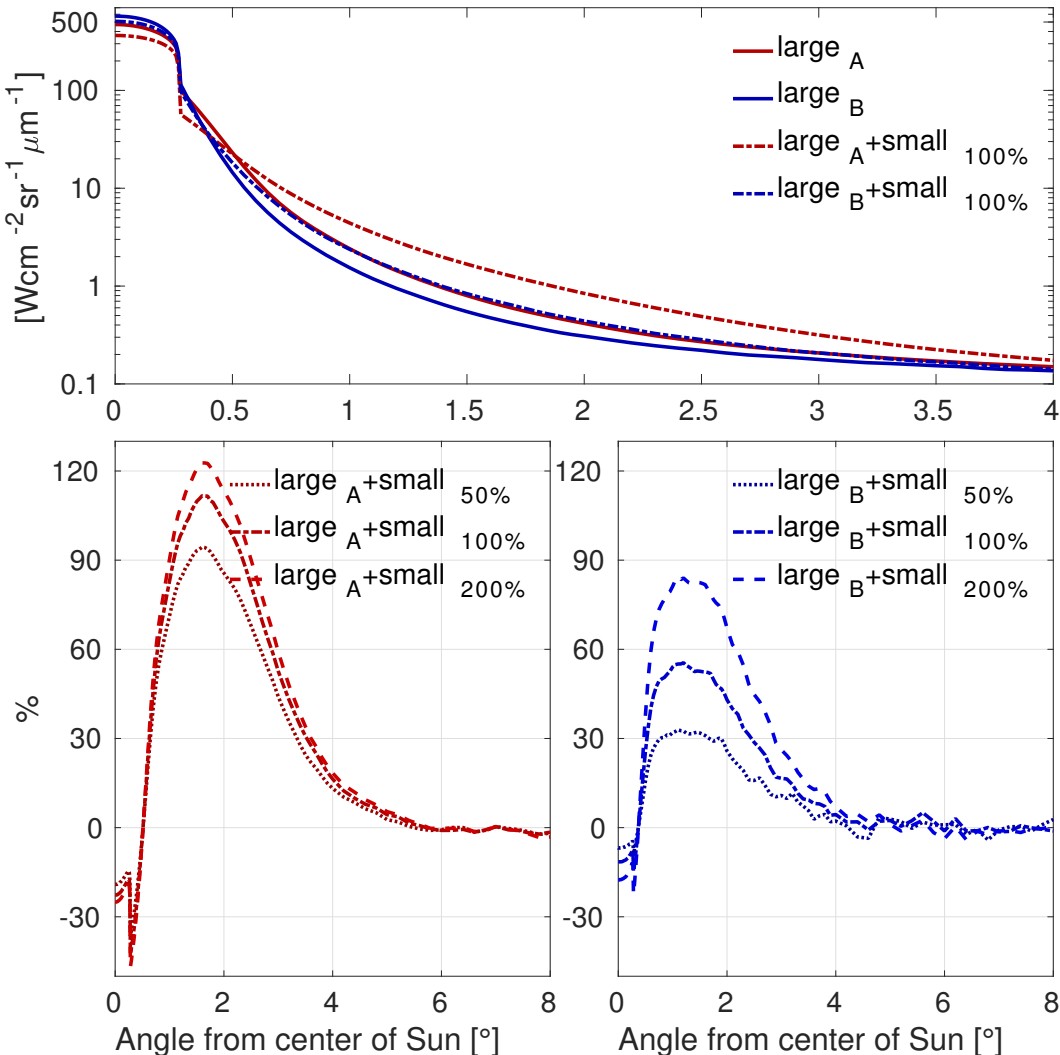

**Figure 9. (a)** Impact of the concentration of small ice crystals on the disk and circumsolar radiances. The simulations are made with the $large_A$ and $large_B$ distributions of large ice crystals including 0–200 % of the measured concentration of small ice crystals assumed to be columns. **(b)** and **(c)** Relative differences to the case with no small ice crystals. In these simulations ice crystals are severely rough, $\tau_a = 0.09$ and $\tau_c = 1.6$.

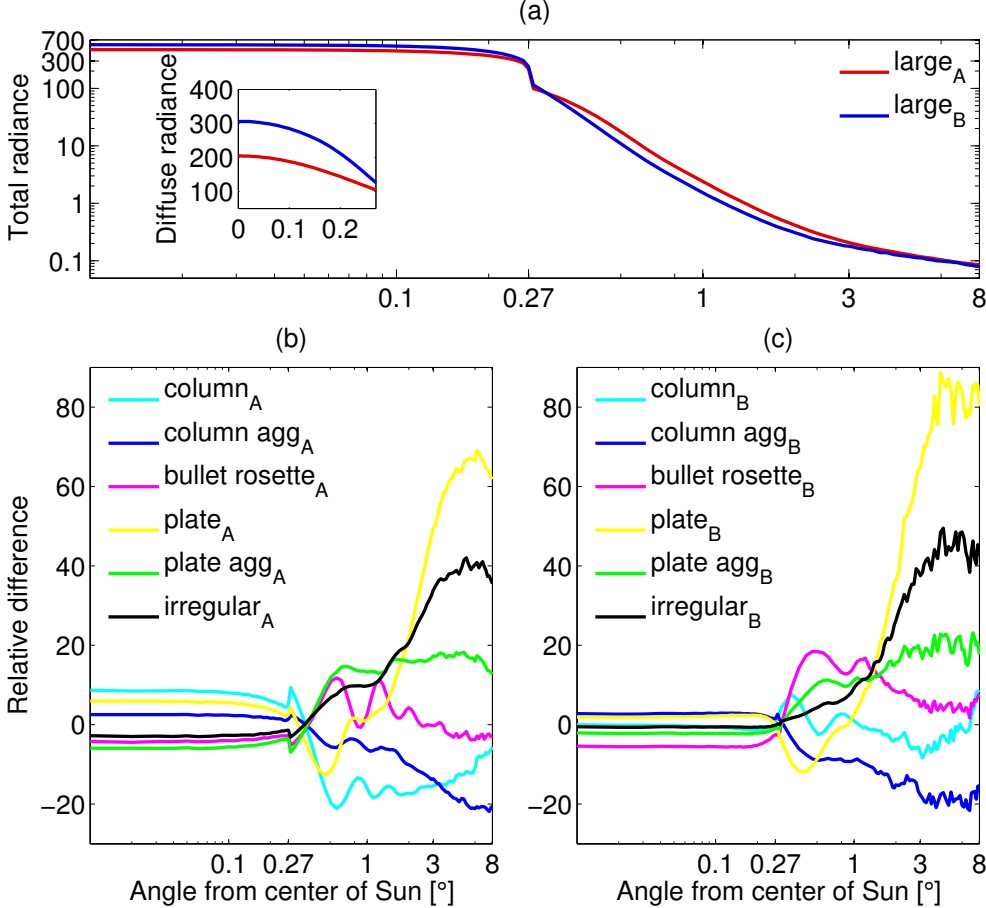

**Figure 10.** Impact of the shape of large severely rough ice crystals on the disk and circumsolar radiances. **(a)** The total radiances based on the $large_A$ and $large_B$ distributions. **(b)** and **(c)** The relative differences of the radiances based on the six single-habit distributions of flight A or B and the $large_A$ or $large_B$, respectively. All the simulations are conducted with $\tau_a = 0.09$ and $\tau_c = 1.6$.

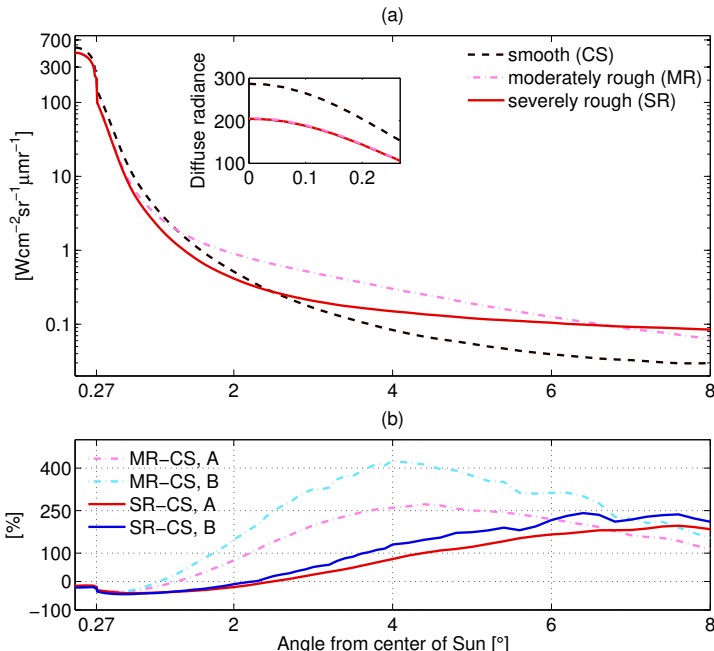

**Figure 11. (a)** Impact of the roughness (smooth, CS, moderately, MR, and severely rough, SR) of large ice crystals on the disk and circumsolar radiances in case of the $large_A$ size-shape distribution. **(b)** Relative differences between results based on the MR or SR and CS ice crystals for the $large_A$ and $large_B$ distributions. In these simulations $\tau_a = 0.09$ and $\tau_c = 1.6$.

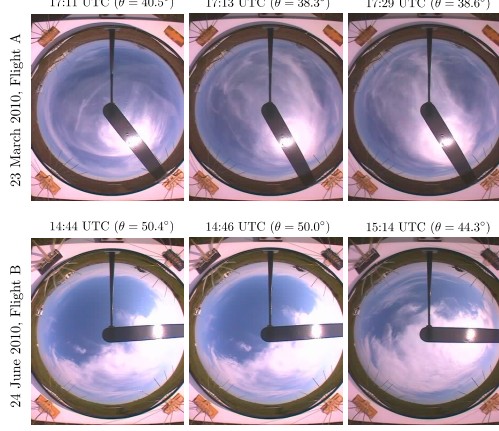

**Figure 12.** Cloud scenes for the six times corresponding to the times of the Sun and Aureole Measurements (SAM) used in the comparison with simulations (see Tables 4 and 5).

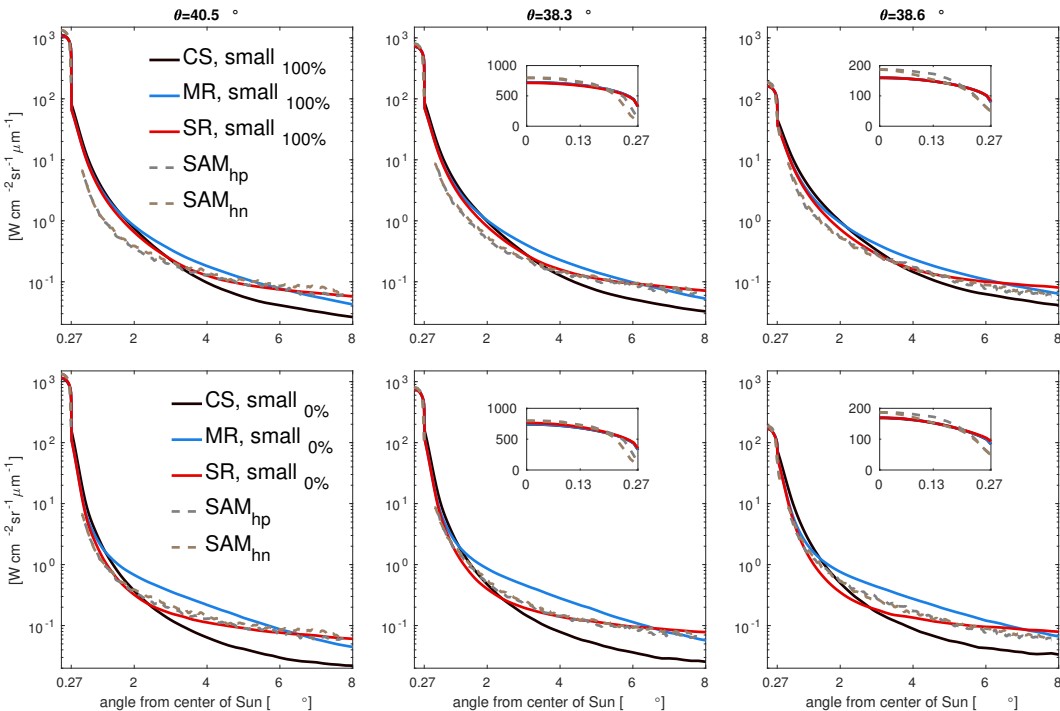

**Figure 13.** Comparison of the Sun and Aureole measured (SAM; hp and hn for the horizontal profiles to the right and left from the center of the sun) and simulated radiances at three measurement times during flight A. For the simulations, the $large_A$ distribution with 100 % and 0 % of measured concentration of small ice crystals is used with $\tau$ and $\theta$ values listed in Table 4. Smooth (CS) and rough (MR and SR) ice crystals are considered.

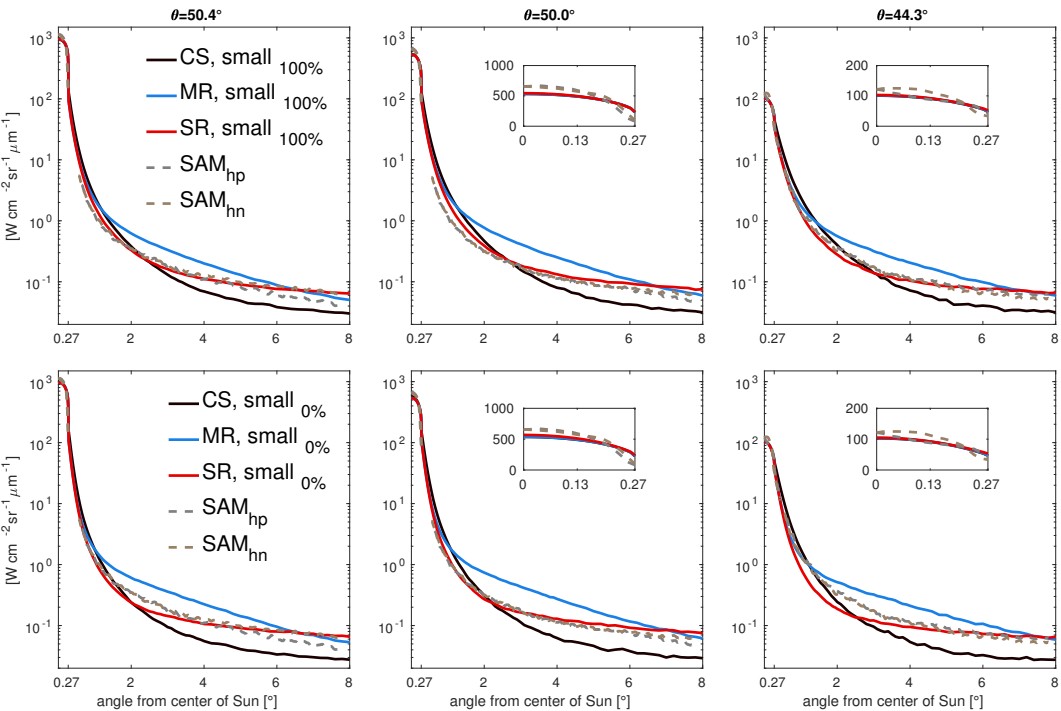

**Figure 14.** Comparison of the Sun and Aureole measured (SAM; hp and hn for the horizontal profiles to the right and left from the center of the sun) and simulated radiances at three measurement times during flight B. For the simulations, the $large_B$ distribution with 100 % and 0 % of measured concentration of small ice crystals is used with $\tau$ and $\theta$ values listed in Table 5. Smooth (CS) and rough (MR and SR) ice crystals are considered.