# Peer review of "Disk and circumsolar radiances in the presence of ice clouds"

_Atmospheric Chemistry and Physics, 2016_

## Referee Comment (RC1) · Anonymous Referee #2 · 23 Dec 2016

The paper deals with the sensitivity analysis of how ice crystals, their shape and roughness, affect direct solar measurements, including the assessment of the diffused and circumsolar radiance.

This is topic of great interest, both to the cloud and aerosol retrieval community, especially from direct sun measurements. In addition, this is of interest to applications that are related to energy harvesting from the sun, as it lays the ground for simulations of the direct solar energy under various atmospheric conditions.

In general, the paper is well written and is well describing of the various methods and sensitivity analysis conducted. However, the connection between the sensitivity studies and observations and to the goal of the study is largely missing from the text. Please try to add more content reminding the reader in each of the sections of why and how

the results of the sensitivity studies would be important for solar measurements and for the aerosol/ice cloud retrieval community.

There are some areas where additional physical explanation or delineation might have been useful. For example, in Fig. 5 and Fig. 9 is it not entirely clear from the text why does the MR crystals result in a much larger bias from CS when compared to the SR particles. One might think that it should result in discrepancies that lie between CS and SR (in magnitude). This might be due to the contradicting effects of the direct and diffuse components, which might create this deflection point, but this is not entirely clear from the text.

In Fig. 11 and 12 it is unclear why some of the SAM measurements (dashed grey lines-hp) are discontinued and showing a drop in radiance intensity around 0.27, while the hn are not. Also, please add the acronym of hp and hn to the figures captions (in Fig.11 and 12), to help the reader.

---

## Referee Comment (RC2) · Anonymous Referee #1 · 1 Jan 2017

The authors studied the relationship between ice cloud particle properties and circumsolar radiance profiles. This topic is of interest for, amongst others, the cloud remote sensing community and may ultimately lead to a better understanding of ice could micro-physics. The authors built their study around rather precious in-situ measurements of cloud micro-physics and simultaneous ground based measurements of circumsolar radiation. For the simulation of radiance profiles they expanded the MC-UniK Monte Carlo radiative transfer model so it can treat the sun as a realistic disc source instead of a point source. Overall the study is therefore within the scope of ACP.

For the most part the paper is technically well written. The authors put their work in reference to previous studies and it is easy to follow which steps they have undertaken in their study. However, what is lacking is a clear formulation of the study goal. Consequently also the presentation of the findings is somewhat vague. Before publication in

ACP these issues should be addressed.

Basically there are two parts to the study: In the first part the authors investigate in sensitivity test the general influence of ice cloud micro-physics on the circumsolar radiance profile. As basis for their tests they use ice particle size-shape distributions measured in-situ at two different days. The sensitivity test are performed using the radiative transfer model. In the second part they investigate under which assumptions the measured radiance profiles can be replicated best. For this they also use the measured size-shape distributions as input for radiative transfer simulations.

While for the second part of the study it makes sense to use only the size-shape distributions measured at the same dates as the radiance profiles, it is unclear why the authors have limited themselves to also only using the two size distributions as basis in the first part of the study. Unfortunately, little information is provided on how representative these size distributions are or whether it is sufficient to focus only on these two size distributions when deriving general relations between ice cloud micro-physics and circumsolar radiation. The authors discuss differences in simulated radiance profiles caused by the differences in the two measured particle distributions as well as due to impact of the assumed particle roughness. However, it remains unclear why the authors did not explore the parameter space further – e.g. by using more size-shape distributions from the SPARTACUS campaign or idealized single-shape size distributions in different size variations. Although certainly not easy to quantify, at least some comment on how common/representative the measured size-shape distributions are considered by the authors should be provided.

Overall the study explores the sensitivity of the phase function in regard to particle shape and roughness. The finding is that the surface roughness is the dominating parameter. The third parameter, particle size, is largely neglected, however. While radiance profiles for three different concentrations of "small particles" are compared, little information is provided about the size distribution(s) used for this small particle fraction. Modifications to the size distribution of the large fraction are not performed.

Following modifications to the script could help to address the above mentioned issues:

- The authors should leave no doubt in the introduction as to what the study goals are.

- The authors should concisely summarize (if deemed feasible maybe also in tabular form for ease of comprehension) which aspects of the radiance profile are influenced by which of the cloud micro-physics parameters. The authors should also mention which of these aspects were newly identified in this study.

- While the authors found that the small particle fraction (Dmax < 100mu) cannot be neglected, the influence of the overall particle size distribution is not very thoroughly explored. I suggest to expand the study in this regard. Alternatively the authors should comment on why they deem the size distribution not to be as important as the particle shape and roughness.

- While only two dates of the SPARTICUS campaign where usable for a comparison to SAM measurements, the authors should add a paragraph that puts the shape-size distributions measured during those two flights in perspective to what was measured during the rest of the campaign.

- The authors should provide the size distribution for the small particle fraction for flights A and B as well as the optical thickness assigned to this particle fraction. The latter could be added to tables 3 and 4.

Additional minor suggestions:

- In line 24 it is stated that circumsolar radiation is caused by scattering on particles between 1mu and 100mu. However, the study mainly focuses on particles larger 100mu. Please clarify/rephrase.

- Should pictures of the cloud scenes (e.g. webcam) for times of comparison between simulations and SAM measurements be available, I suggest to include those.

- Caption of Figure 5: "Sensitivity of the size and vertically integrated phase functions to the roughness of large ice crystals.". Potentially remove "the size" from the sentence.

---

## Author Comment (AC1) · 4 Mar 2017

We thank Anonymous Referee #2 for his/her constructive and insightful comments on the manuscript. Below, we respond to these comments and outline changes planned in the revised manuscript.
* * *
COMMENT:

. . . In general, the paper is well written and is well describing of the various methods and sensitivity analysis conducted. However, the connection between the sensitivity studies and observations and to the goal of the study is largely missing from the text. Please try to add more content reminding the reader in each of the sections of why and

how the results of the sensitivity studies would be important for solar measurements and for the aerosol/ice cloud retrieval community.

RESPONSE:

The overarching goal of the research is to understand how ice clouds influence the downwelling solar radiances within a few degrees from the direction of the Sun. This knowledge may be exploited, in future work, for developing schemes to correct measurements of direct solar radiation for the diffuse radiation that is present at the angular range of instruments such as pyrheliometers. Furthermore, it is crucial for understanding the information content in measurements with the relatively new SAM instrument, and for the future development of retrieval algorithms based on SAM data. As noted by both reviewers, this study is largely divided into two components, both of which contribute to the overarching goal. The goal of the first component (i.e., the sensitivity studies) is to determine what parameters the circumsolar radiance is sensitive to; the goal of the second component is to use case studies to determine if we are able to get a successful match between the observed and simulated radiances. However, while we consider the first component (sensitivity studies) interesting in its own right, it also provides important information for designing and interpreting the comparison with modelled radiances. Specifically, it demonstrates the large sensitivity of circumsolar radiances to ice crystal roughness and small ice particles. This, together the fact that in-situ microphysical measurements yield no information on roughness and only very uncertain information on small ice crystals, motivates the study of how assumptions related to these factors impact the agreement between modelled and measured radiances. In the revised manuscript, we will clarify the goals of the study in the Introduction. In addition, the links between the sensitivity tests and the comparison between observations (noted above) will be made explicit in a new subsection (section 4.4 in the revised manuscript) summarizing the findings of the sensitivity tests. Regarding, the last suggestion ("add more content reminding the reader in each of the sections of why and how the results of the sensitivity studies would be important for solar measurements and for the aerosol/ice cloud retrieval community"), we think this discussion is best added to the new subsection summarizing the sensitivity tests, rather than in each subsection separately.
* * *
COMMENT:

There are some areas where additional physical explanation or delineation might have been useful. For example, in Fig. 5 and Fig. 9 is it not entirely clear from the text why does the MR crystals result in a much larger bias from CS when compared to the SR particles. One might think that it should result in discrepancies that lie between CS and SR (in magnitude). This might be due to the contradicting effects of the direct and diffuse components, which might create this deflection point, but this is not entirely clear from the text.

RESPONSE:

This was already explained in the original manuscript (lines 307-316) in connection to phase functions, but apparently, the explanation was not sufficiently clear. This issue is related to how the treatment of ice crystal roughness impacts the paths of rays transmitted through parallel crystal faces. In the case of completely smooth crystals, such ray paths results in (near-) delta-transmission, increasing the phase function at scattering angles very close to zero. However, in the case of "rough" crystals, the ice crystal surface slopes are distorted randomly for each incident ray. In effect, this eliminates ray paths that pass through exactly parallel faces. This is why for both moderately rough (MR) and severely rough (SR) crystals, the phase function is lower than that for completely smooth crystals in very-near-forward scattering directions. Moreover, since virtually all such ray paths are eliminated both in the case of MR and SR crystals, the phase function for MR and SR crystals is nearly identical at angles smaller than about $0.3°$ in Fig. 5. That is, the same amount of energy is removed from the very-near-forward scattering for both MR and SR crystals, and added at larger scattering angles.

In the case of MR crystals, most of this energy is distributed within a few degrees from the forward direction, while for the SR crystals, it is distributed over a larger range of scattering angles. Therefore, in the case of MR crystals, the phase function is larger than for SR crystals at relatively small scattering angles (up to about $6°$), and smaller at larger scattering angles. Of course, these arguments also apply to the radiances.

We will try to make this issue "crystal clear" in the revised manuscript.
* * *
COMMENT:

In Fig. 11 and 12 it is unclear why some of the SAM measurements (dashed grey lines-hp) are discontinued and showing a drop in radiance intensity around 0.27, while the hn are not. Also, please add the acronym of hp and hn to the figures captions (in Fig.11 and 12), to help the reader.

RESPONSE:

The vertical drop in hp lines in Figures 11 and 12 is present because of the gaps between the measurements from the solar disk and the solar aureole cameras in the SAM instrument. The inner edge of the solar aureole imagery in the SAM 300 model in use at the ARM SGP site during SPARTACUS is at about $0.6°$ from the centre of the solar disk. When the optical depth is below some value, which depends upon the degree of forward directivity of the scattering phase functions, the outer edge of the aureole in the solar disk camera falls below its intensity threshold and creates a gap when the two images are merged (as seen in Figures 11 and 12). These gaps should exist in both the positive and negative, horizontal and vertical profiles, unless the optical thickness along the line of sight has attenuated the disk radiance sufficiently that both the disk and aureole portions of the radiance in the gap are within the intensity measurement ranges of the respective cameras. In the SAM data shown in Figures 11 and 12 the gap is present in all hn and hp lines even it is hard to distinguish from the

figures. For example in the rightmost panels of Figures 11 and 12 ($\theta$=38.6°; $\tau$=2.1 and $\theta$=44.3°; $\tau$=2.3) the gap is at $0.36°-0.52°$ and $0.46°-0.52°$ from the centre of the Sun, respectively. In the revised manuscript the meaning of acronyms hp and hn will be stated more clearly in the figure captions of Figures 11 and 12. In addition, we will remove the unphysical parts of the hp lines (i.e. the near-vertical drop/increase) from the shown SAM measurements and explain in the plain text the reason for the gap in the SAM measurements.

---

## Author Comment (AC2) · 4 Mar 2017

We thank Anonymous Referee #1 for his/her constructive and insightful comments on the manuscript. Below, we respond to these comments and outline changes planned in the revised manuscript.
* * *
COMMENT:

For the most part the paper is technically well written. The authors put their work in reference to previous studies and it is easy to follow which steps they have undertaken in their study. However, what is lacking is a clear formulation of the study goal. Consequently also the presentation of the findings is somewhat vague. Before publication in

ACP these issues should be addressed.

RESPONSE:

The overarching goal of the research is to understand how ice clouds influence the downwelling solar radiances within a few degrees from the direction of the Sun. This knowledge may be exploited, in future work, for developing schemes to correct measurements of direct solar radiation for the diffuse radiation that is present at the angular range of instruments such as pyrheliometers. Furthermore, it is crucial for understanding the information content in measurements with the relatively new SAM instrument, and for the future development of retrieval algorithms based on SAM data.

As noted by both reviewers, this study is largely divided into two components, both of which contribute to the overarching goal. The goal of the first component (i.e., the sensitivity studies) is to determine what parameters the circumsolar radiance is sensitive to; the goal of the second component is to use case studies to determine if we are able to get a successful match between the observed and simulated radiances. However, while we consider the first component (sensitivity studies) interesting in its own right, it also provides important information for designing and interpreting the comparison with modelled radiances. Specifically, it demonstrates the large sensitivity of circumsolar radiances to ice crystal roughness and small ice particles. This, together the fact that in-situ microphysical measurements yield no information on roughness and only very uncertain information on small ice crystals, motivates the study of how assumptions related to these factors impact the agreement between modelled and measured radiances.

In the revised manuscript, we will clarify the goals of the study in the Introduction. In addition, the links between the sensitivity tests and the comparison between observations (noted above) will be made explicit in a new subsection (section 4.4 in the revised manuscript) summarizing the findings of the sensitivity tests.
* * *
COMMENT:

While for the second part of the study it makes sense to use only the size-shape distributions measured at the same dates as the radiance profiles, it is unclear why the authors have limited themselves to also only using the two size distributions as basis in the first part of the study. Unfortunately, little information is provided on how representative these size distributions are or whether it is sufficient to focus only on these two size distributions when deriving general relations between ice cloud micro-physics and circumsolar radiation. The authors discuss differences in simulated radiance profiles caused by the differences in the two measured particle distributions as well as due to impact of the assumed particle roughness. However, it remains unclear why the authors did not explore the parameter space further – e.g. by using more size-shape distributions from the SPARTACUS campaign or idealized single-shape size distributions in different size variations. Although certainly not easy to quantify, at least some comment on how common/representative the measured size-shape distributions are considered by the authors should be provided.

Overall the study explores the sensitivity of the phase function in regard to particle shape and roughness. The finding is that the surface roughness is the dominating parameter. The third parameter, particle size, is largely neglected, however. While radiance profiles for three different concentrations of "small particles" are compared, little information is provided about the size distribution(s) used for this small particle fraction. Modifications to the size distribution of the large fraction are not performed.

RESPONSE:

In the revised manuscript, sensitivity tests will be added which demonstrate the impact of ice crystal size through the use of idealized (lognormal) size distributions. These tests demonstrate that the impact of ice crystal size arises, to a large part, through its impact on the width of the diffraction peak. In broad terms, the diffuse radiance in the solar disk area increases, and the radiance outside the solar disk decreases with

increasing ice crystal size, while at angles of more than a few degrees the effect of ice crystal size is relatively small, especially for rough ice crystals. In addition, in the revised manuscript Figure 2 will also include the measured size distributions of small ice crystals.
* * *
COMMENT:

Following modifications to the script could help to address the above mentioned issues:

The authors should leave no doubt in the introduction as to what the study goals are.

RESPONSE:

The goals will be clarified, as stated in our response to the first general comment.
* * *
COMMENT:

The authors should concisely summarize (if deemed feasible maybe also in tabular form for ease of comprehension) which aspects of the radiance profile are influenced by which of the cloud micro-physics parameters. The authors should also mention which of these aspects were newly identified in this study.

RESPONSE:

In the revised manuscript, a subsection summarizing the main findings of the sensitivity tests, and how they relate to earlier research, will be added (section 4.4).
* * *
COMMENT:

While the authors found that the small particle fraction (Dmax < 100mu) cannot be neglected, the influence of the overall particle size distribution is not very thoroughly

explored. I suggest to expand the study in this regard. Alternatively the authors should comment on why they deem the size distribution not to be as important as the particle shape and roughness

RESPONSE:

In the revised manuscript, sensitivity tests will be added which demonstrate the impact of ice crystal size through the use of idealized (lognormal) size distributions (see above).
* * *
COMMENT

While only two dates of the SPARTICUS campaign where usable for a comparison to SAM measurements, the authors should add a paragraph that puts the shape-size distributions measured during those two flights in perspective to what was measured during the rest of the campaign.

RESPONSE:

This will be done in the revised manuscript. Jackson et al. (2015) examined the size and shape distributions sampled by the SPEC Learjet for all 101 missions flown during SPARTICUS, establishing the meteorological context of each cirrus sampled using visible and infrared images from GOES and WSR 88D radar images. Comparing Figure 2 of the manuscript (size-shape distributions of flight A and B) against Figure 10 in Jackson et al. (2015) establishes the degree to which the data from these 2 flights were representative of those observed during other flights: flight A tends to have lower N(D) than the average observed during other flights whereas flight B tends to have larger N(D) than the observed averages. Overall, flights A and B well represent the range of conditions observed during SPARTICUS.
* * *
[Figure]

COMMENT:

The authors should provide the size distribution for the small particle fraction for flights A and B as well as the optical thickness assigned to this particle fraction. The latter could be added to tables 3 and 4.

RESPONSE:

This information will be presented in the revised manuscript.
* * *
COMMENT:

Additional minor suggestions:

In line 24 it is stated that circumsolar radiation is caused by scattering on particles between 1mu and 100mu. However, the study mainly focuses on particles larger 100mu. Please clarify/rephrase.

RESPONSE:

Good point. Indeed, the upper limit is not warranted, because circumsolar radiation is particularly large in the presence of particles much larger than the wavelength. In the revised manuscript, this sentence will be modified as follows: "The radiation arises from near-forward scattering of direct solar radiation by atmospheric particles with sizes comparable to or larger than the wavelength (i.e., larger than about 1 $\mu$m)."
* * *
COMMENT:

Should pictures of the cloud scenes (e.g. webcam) for times of comparison between simulations and SAM measurements be available, I suggest to include those.

RESPONSE:

[Figure]

Total Sky Imager (TSI) images for times corresponding to those of SAM measurements used in the comparison with simulations will be included in the revised manuscript.
* * *
COMMENT:

Caption of Figure 5: "Sensitivity of the size and vertically integrated phase functions to the roughness of large ice crystals.". Potentially remove "the size" from the sentence.

RESPONSE:

In the revised manuscript, the words "the size" will be removed from this sentence and also from the sentence in the caption of Figure 4.

---

## Author Response (AR1)

**Response to the reviewer comments**

**Manuscript number: acp-2016-967**

Manuscript title: "Disk and circumsolar radiances in the presence of ice clouds"

\*\*\*\*\*\*\*\*\*\*\*

**A note on the changes in the structure of the manuscript.**

• The order of original sections 4.2 and 4.3 has been reversed, as we thought that it is actually more logical to go through the impact of size on radiances before the impact of ice crystal shape and roughness. Also, these sections have been renamed. The original section 4.3 ("Sensitivity of the radiances to small ice crystals") is now section 4.2 "Sensitivity of radiances to ice crystal size", and includes an additional test of lognormal size distributions, in response to a comment by Referee #1. The original section 4.2 ("Sensitivity of the radiances to properties of large ice crystals") is now Section 4.3: "Sensitivity of the radiances to the shape and roughness of ice crystals".

• A new subsection (Section 4.4: Summary of sensitivity tests) has been added in response to the Referee comments.

• Two new figures (Figs 5 and 12) have been added in response to the Referee comments.

**Response to comments by Referee #1**

We thank Anonymous Referee #1 for his/her constructive and insightful comments on the manuscript. Below, we respond to these comments and outline the made changes in the revised manuscript.

**COMMENT:**

For the most part the paper is technically well written. The authors put their work in reference to previous studies and it is easy to follow which steps they have

undertaken in their study. However, what is lacking is a clear formulation of the study goal. Consequently also the presentation of the findings is somewhat vague. Before publication in ACP these issues should be addressed.

**RESPONSE:**

In the revised manuscript, we have clarified the goals of the study in the Introduction section.

**CHANGE IN MANUSCRIPT:**

The study goals are now formulated more clearly in lines 83-92 and 104-107 of the revised manuscript.

**COMMENT:**

While for the second part of the study it makes sense to use only the size-shape distributions measured at the same dates as the radiance profiles, it is unclear why the authors have limited themselves to also only using the two size distributions as basis in the first part of the study. Unfortunately, little information is provided on how representative these size distributions are or whether it is sufficient to focus only on these two size distributions when deriving general relations between ice cloud microphysics and circumsolar radiation. The authors discuss differences in simulated radiance profiles caused by the differences in the two measured particle distributions as well as due to impact of the assumed particle roughness. However, it remains unclear why the authors did not explore the parameter space further – e.g. by using more size-shape distributions from the SPARTACUS campaign or idealized single-shape size distributions in different size variations. Although certainly not easy to quantify, at least some comment on how common/representative the measured size-shape distributions are considered by the authors should be provided.

Overall the study explores the sensitivity of the phase function in regard to particle shape and roughness. The finding is that the surface roughness is the dominating parameter. The third parameter, particle size, is largely neglected, however. While radiance profiles for three different concentrations of "small particles" are compared, little information is provided about the size distribution(s) used for this small particle fraction. Modifications to the size distribution of the large fraction are not performed.

**RESPONSE:**

In the revised manuscript we have added sensitivity tests which demonstrate the impact of ice crystal size through the use of idealized (lognormal) size distributions. These tests demonstrate that the impact of ice crystal size arises, to a large part, through its impact on the width of the diffraction peak. In broad terms, the diffuse radiance in the solar disk area increases, and the radiance outside the solar disk decreases with increasing ice crystal size, while at angles of more than a few degrees the effect of ice crystal size is relatively small, especially for rough ice crystals. In addition, Figure 2 now also includes the measured size distributions of small ice crystals.

**CHANGE IN MANUSCRIPT:**

The new lognormal size-shape distributions and their optical properties are explained and described in the revised manuscript in lines 248-254, 298-315. Figure 4 is modified to account P11 of the lognormal size distributions. New discussion on the impact of size on the angular distribution of radiances is included in Section 4.2 (lines 398-410) of the revised manuscript. As noted in the beginning of this response letter, the order of Sections 4.2 and 4.3 has been changed. The impact of size is also discussed in Section 4.4 (Table 3 and lines 485-500). Conclusions related to size are modified in Section 6 (lines 601-607).

The vertically averaged size distributions for flight A and B including crystals smaller than D

1Department of Physics, University of Helsinki, P.O.Box 48, FI-00014 University of Helsinki, Finland

2Finnish Meteorological Institute, P.O.Box 503, FI-00101 Helsinki, Finland

3Department of Atmospheric Science, University of Illinois at Urbana-Champaign, Urbana, 105 S. Gregory ST., IL 61801-3070, USA

4Leibniz Institute for Tropospheric Research, Permoserstraße 15, 04318 Leipzig, Germany
 5Research Department, Swedish Meteorological and Hydrological Institute, Folkborgsvägen 17, 601 76 Norrköping, Sweden

6Department of Earth and Space Science, Chalmers University of Technology, 412 96 Gothenburg, Sweden

7Visidyne, Inc., 429 Stanley Drive, Santa Barbara, CA 93105, USA

Correspondence to: Päivi Haapanala (paivi.haapanala@helsinki.fi)

**Abstract.** The impact of ice clouds on solar-disk and circumsolar radiances is investigated using a Monte Carlo radiative transfer model. The monochromatic direct and diffuse radiances are simulated at angles of 0 o-to 8° from the center of the Sunsun. Input data for the model are derived from measurements conducted during the 2010 Small Particles in Cirrus (SPARTICUS) campaign together

5 with state-of-the-art databases of optical properties of ice crystals and aerosols. For selected cases, the simulated radiances are compared with ground-based radiance measurements with obtained by the Sun and Aureole Measurement (SAM) instrument.

First, the sensitivity of the radiances to the ice cloud properties and aerosol optical thickness was is addressed. The angular dependence of the disk and circumsolar radiances was is found to be most

- 10 sensitive to assumptions about ice crystal roughness (or, more generally, non-ideal features of ice crystals) and size distribution, with ice crystal habit playing a somewhat smaller role. Second, in the comparisons with SAM data, the ice-cloud optical thickness was is adjusted for each case so that the simulated radiances agreed agree closely (i.e., within 3 %) with the measured disk radiances. Circumsolar radiances at angles larger than  $\approx 3^{\circ}$  were are systematically underestimated when as-
- 15 suming smooth ice crystals, but whereas the agreement with the measurements was is better when rough ice crystals were 
[revised manuscript text omitted]
                                                   | $\underbrace{\operatorname{dir} 0.0^{\circ}}_{\leftarrow} \underbrace{-0.27^{\circ}}_{\leftarrow}$ | $\underbrace{\text{diff } 0.0^{\circ}-0.27^{\circ}}_{\leftarrow\leftarrow\leftarrow\leftarrow}$ | $\underbrace{\text{diff} \ 0.27}_{\sim}^{\circ} \underbrace{-1}_{\sim}^{\circ}$ | $\underbrace{\text{diff.}1^{\circ}}_{==}3^{\circ}$ | $\underbrace{\text{diff.} 3^{\circ} - 8^{\circ}}_{\sim \sim}$ |
|-------------------------------------------------------------|----------------------------------------------------------------------------------------------------|-------------------------------------------------------------------------------------------------|---------------------------------------------------------------------------------|----------------------------------------------------|---------------------------------------------------------------|
| optical thickness ( $\tau_c$ =1.6)                          | ****                                                                                               | +~                                                                                              | t                                                                        | +~                                                 | +~~                                                           |
| $\underline{\text{median size } (D_0 = 200 \ \mu\text{m})}$ | ~~                                                                                                 | t                                                                                               | ttt                                                                             | * ++ +                                      | *~                                                            |
| small crystals $D < 100 \mu m (large_{A/B})$                | ~~                                                                                                 | t                                                                                        | t                                                                        | ++ ~                                        | + ~                                                    |
| shape distribution $(large_{A/B})_{\sim}$                   | ~~                                                                                                 | $\bar{\sim}$                                                                                    | ~                                                                               | + ~                                         | + ~                                                    |
| roughness (CS)                                              |                                                                               | <del>++++</del> ~  | ****                                                          |

**Table 4.** The values of solar zenith angle  $\theta$  and optical thickness of cloud ( $\tau_c$ ), aerosols ( $\tau_a$ ), and gases ( $\tau_{gases}$ ) used in the comparison simulations for flight A. The cloud is described with the size-shape distributions  $large_A$  and  $large_A + small_{100\%}$  of rough (MR and SR) and completely smooth (CS) ice crystals. Values The fractional contribution of small ice crystals to cloud optical thickness for the  $large_A + small_{100\%}$  size-shape distribution ( $f_{small}$ ) and the total optical thickness (cloud+aerosols) retrieved from the Sun and Aureole measurements (SAM) are also shownlisted.

| $\theta$ [°]                                 | 40.5        | 38.3        | 38.6        |
|----------------------------------------------|-------------|-------------|-------------|
| $	au_{gases}$                                | 0.072       | 0.072       | 0.072       |
| $\tau_a$ (AERONET, MFRSR)                    | 0.09        | 0.09        | 0.09        |
| $f_{small}, large_A + small_{100\%}$         | 79 % | 79 % | 79 % |
| $\tau_c$ , CS, $large_A + small_{100\%}$     | 0.6         | 1.05        | 2.5         |
| $\tau_c$ , MR/SR, $large_A + small_{100\%}$  | 0.6         | 1.0         | 2.4         |
| $\tau_c$ , CS, $large_A + small_{0\%}$       | 0.75        | 1.25        | 3.1         |
| $	au_c, \text{MR/SR}, large_A + small_{0\%}$ | 0.65        | 1.15        | 2.75        |
| $	au_{SAM}$                                  | 0.6         | 1.0         | 2.1         |

**Table 5.** The values of solar zenith angle  $\theta$  and optical thickness of cloud ( $\tau_c$ ), aerosol ( $\tau_a$ ) - and gases ( $\tau_{gases}$ ) used in the comparison simulations for flight B. The cloud is described with the size-shape distributions  $large_B$  and  $large_B + small_{100\%}$  of rough (MR and SR) and completely smooth (CS) ice crystals. Values The fractional contribution of small ice crystals to cloud optical thickness for the  $large_B + small_{100\%}$  size-shape distribution ( $f_{small}$ ) and the total optical thickness (cloud+aerosols) retrieved from the Sun and Aureole measurements (SAM) are also shownlisted.

| $\theta$ [°]                                | 50.4  | 50.0        | 44.3        |
|---------------------------------------------|-------|-------------|-------------|
| $	au_{gases}$                               | 0.074 | 0.074       | 0.074       |
| $\tau_a$ (AERONET, MFRSR)                   | 0.166 | 0.166       | 0.166       |
| $f_{small}, large_B + small_{100\%}$        | 27 %  | 27 % | 27 % |
| $\tau_c$ , CS, $large_B + small_{100\%}$    | 0.7   | 1.3         | 3.5         |
| $\tau_c$ , MR/SR, $large_B + small_{100\%}$ | 0.6   | 1.15        | 3.05        |
| $\tau_c, CS, large_B + small_{0\%}$         | 0.75  | 1.45        | 4.0         |
| $\tau_c$ , MR/SR, $large_B + small_{0\%}$   | 0.65  | 1.25        | 3.3         |
| $	au_{SAM}$                                 | 0.6   | 1.0         | 2.3         |

**Figure 1.** Detector positions in the MC-UniK model cover angles from 0 to -8 and 8° from the center of the Sun  $\underbrace{\text{sun}}_{\text{vn}}(0^\circ, 0^\circ)$ . Both horizontal and vertical cross sections are divided into positive and negative parts (hp; hn; vp; vn). The circle demonstrates the size of the solar disk, with a diameter of  $0.534^\circ$ .